# E3 Ubiquitin Ligase TRIP12: Regulation, Structure, and Physiopathological Functions

**DOI:** 10.3390/ijms21228515

**Published:** 2020-11-12

**Authors:** Manon Brunet, Claire Vargas, Dorian Larrieu, Jérôme Torrisani, Marlène Dufresne

**Affiliations:** 1Institut National de la Santé et de la Recherche Médicale, INSERM Unit 1037, Centre de Recherches en Cancérologie de Toulouse, CEDEX 1, 31 037 Toulouse, France; manon.brunet@inserm.fr (M.B.); claire.vargas@inserm.fr (C.V.); dorian.larrieu@gmail.com (D.L.); 2Université Toulouse III-Paul Sabatier, CEDEX 9, 31 062 Toulouse, France

**Keywords:** TRIP12, E3 ubiquitin ligase, cancers, intellectual disorders

## Abstract

The Thyroid hormone Receptor Interacting Protein 12 (TRIP12) protein belongs to the 28-member Homologous to the E6-AP C-Terminus (HECT) E3 ubiquitin ligase family. First described as an interactor of the thyroid hormone receptor, TRIP12’s biological importance was revealed by the embryonic lethality of a murine model bearing an inactivating mutation in the *TRIP12* gene. Further studies showed the participation of TRIP12 in the regulation of major biological processes such as cell cycle progression, DNA damage repair, chromatin remodeling, and cell differentiation by an ubiquitination-mediated degradation of key protein substrates. Moreover, alterations of TRIP12 expression have been reported in cancers that can serve as predictive markers of therapeutic response. The *TRIP12* gene is also referenced as a causative gene associated to intellectual disorders such as Clark–Baraitser syndrome and is clearly implicated in Autism Spectrum Disorder. The aim of the review is to provide an exhaustive and integrated overview of the different aspects of TRIP12 ranging from its regulation, molecular functions and physio-pathological implications.

## 1. Introduction

Thyroid hormone Receptor Interacting Protein 12 cDNA was cloned from the human myeloid KG-1 cell line and named the unidentified KIAA0045 human gene. The deduced coding sequence of KIAA0045 showed partial identity to that of the *S. cerevisiae* UFD4 protein (Ubiquitin Fusion Degradation 4) [1]. Another study identified Thyroid hormone Receptor Interacting Protein 12 (TRIP12) as a member of the structurally and functionally related E3 ubiquitin ligases based on the identification of a domain homologous to the E6-associated protein carboxyl terminus (HECT domain), which is a protein that induces the ubiquitin-dependent degradation of P53 in the presence of the E6 protein from the papillomavirus [2].

TRIP12 was later characterized as a thyroid hormone receptor-interacting protein by the trap yeast interaction system to isolate proteins that interact with the ligand binding/dimerization/transcriptional activation domain of the rat TR-β1 (Thyroid Receptor-β1) [3]. Fifteen cDNAs encoding Thyroid hormone Receptor Interacting Proteins (TRIPs) were isolated. All the TRIPs interacted with the Thyroid hormone Receptor (TR), but only when thyroid hormone T3 was present (TRIP1 to TRIP11) or only when T3 was absent (TRIP12 to TRIP15). Protein sequences of the fifteen TRIPs are unrelated, but sequences similarities with known proteins and functional motifs were found for nine of them based on blast alignment analyses with no demonstration of functional significance. The TRIP12 sequence was found to show 55% identity with the C-terminus of E6-associated protein, and more than a decade following its identification, TRIP12 was demonstrated to function as an E3 ubiquitin ligase [4]. Evidence of functional roles was later provided for the other TRIPs proteins (Appendix A). Among them, TRIP12 is the only TRIP family member bearing an E3 ubiquitin ligase activity.

The biological importance of TRIP12 was revealed by the embryonic lethality of a murine model bearing an inactivating mutation in the *TRIP12* gene [5]. Several molecular functions of TRIP12 in important cellular processes and signaling pathways have been demonstrated in recent years. Accumulating evidence indicate that TRIP12 ubiquitinates key proteins for cell homeostasis, regulates gene expression (See Section 5), and plays important roles in cancers and neurological diseases. Up to now, more than 58 publications related to TRIP12 have been referenced in PubMed, half of them in the last 5 years. We have recently contributed to broadening knowledge on the role of TRIP12 by discovering a new substrate essential for pancreatic acinar cells differentiation [6] and by demonstrating that TRIP12 is a new chromatin-associated protein with several implications in the cell cycle progression and in the maintenance of genome integrity [7]. Due to its implication in several important physiological and pathological processes, TRIP12 now emerges as a protein of interest for molecular mechanisms researches and as a potential new therapeutic target.

In this review, we attempt to comprehensively summarize the current knowledge from the literature and databases on TRIP12. This summary includes the gene and protein expression, the protein structure, the protein interactors and functions. We address the physio-pathological roles of TRIP12 in essential biological pathways in relation with some of its substrates. We also present the alterations of the *TRIP12* gene found in neurodevelopmental disorders and cancers as well as the alterations of TRIP12 protein expression in cancers, and their consequences. Finally, we discuss the potential role of TRIP12 as a therapeutic target and highlight the need for further research studies on this protein.

## 2. TRIP12 mRNA and Protein Expression

### 2.1. TRIP12 Gene Organization

In the human genome, the gene encoding for *TRIP12* mRNA is located on the long arm of chromosome 2 at the locus q36.3. It spreads over nearly 168 kb (Genome Browser (see Section Software and databases)). *TRIP12* mRNA expression is driven by a bidirectional promoter that also initiates the transcription of *FBXO36* (F-box Only protein 36) mRNA in a head-to-head opposite direction. Bidirectional promoters are often used to control genes sharing the same regulation or encoding proteins exhibiting a similar biological process [8]. The *FBXO36* gene encodes a protein that belongs to the F-box protein family. F-box is a ≈50 aa-domain that functions as a site for protein–protein interaction. Up to now, the functions of FBXO36 are unknown. In general, F-box proteins are part of SCF (SKP1, Cullin, F-box protein) ubiquitin ligase complexes, in which they bind to substrates for ubiquitin-mediated proteolysis [8,9]. This would be consistent with the fact that TRIP12 and FBXO36 expression are co-regulated. The 555 bp-bidirectional promoter is embedded in a 1753 bp long-CG-rich region also called CpG (Cytosine-phosphate-Guanosine) island, but the regulatory regions and methylation level of *TRIP12/FBXO36* promoter remain unstudied.

### 2.2. TRIP12 mRNA Expression

To date, 26 different *TRIP12* splicing mRNA variants have been identified (www.ncbi.nlm.nih.gov) (Table 1). Their length fluctuates from 9,065 (variant 4) to 10,492 nt (variant 20) with 41 to 43 exons with the exception of variants 24, 25, and 26 that contain only the first six (variants 24 and 25) or seven exons (variant 26) with a length of 5,951 (variant 26) to 6,052 nt (variant 25). Little is known regarding the expression level of the different splicing variants in human cells. It is suggested *TRIP12* mRNA transcripts are a mixture of all possible isoforms, although the variant NM_001284215.1 seems to be predominant in peripheral leukocytes [10]. For the long variants, while the 5′-Untranslated Region (5′-UTR) is ≈180 nt-long, the 3′-Untranslated Region (3′-UTR) represents approximately one-third of the total mRNA length (≈3700 nt), which likely indicates a regulatory function for this region (Figure 1). In fact, the existence of a short (340 nt) and a long form (3716 nt) of *TRIP12* 3′-UTR mRNA was reported [11]. The different sites of transcriptional termination are explained by the presence of two consensus polyadenylation signals (Figure 1). These two variants display different translation efficiency in response to an activation of the mTORC (mammalian Target Of Rapamycin Complex 1) signaling pathway where the short form is more efficiently translated [11]. With reference to the Human Protein Atlas (see Section Software and databases), the expression of *TRIP12* mRNA is ubiquitous in human tissues even if levels of mRNA vary between tissues. For example, *TRIP12* mRNA is highly expressed in testis and in muscles but weakly expressed in pancreatic tissue [1,12,13].

### 2.3. TRIP12 Protein Expression and Cellular Localisation

TRIP12 protein translation can occur from 26 different mRNA splicing variants (Table 1). This leads to the generation of 14 different TRIP12 protein isoforms (isoforms a to m) with a length comprised between 1722 aa and 2068 aa for the long isoforms a to l (Figure 2). The two short isoforms (m and n) contain only 440 and 411 aa, respectively. Multiple alignment analysis COBALT (Constraint-based Multiple Alignment Tool) (see Section Software and databases) reveals a high similarity between the long isoforms (Figure 2). The two most studied isoforms b and c (2025 and 1992 aa, respectively) only differ by a six aa deletion in the IDR domain and a 27 aa deletion in the WWE domain in the isoform c that potentially modifies the protein–protein interaction capacity of this isoform (see Section 3.2.). For further description of the TRIP12 protein, the 1992 aa isoform c (NP_004229.1) will be used as a reference. Even though several protein isoforms of TRIP12 potentially exist in human cells, it is still unknown whether the different isoforms are cell-specific or if they co-exist and in what proportion in the same type of cells. Unfortunately, commercially available antibodies are directed against epitopes that are common to the long isoforms of TRIP12 (except the isoform d). Therefore, they cannot discriminate the different long isoforms. However, immunostaining analysis using these antibodies reveals a nuclear localization of TRIP12 in several cell lines [7,14] and in human tissues (Human Protein Atlas). It can be explained by the presence of two putative bipartite variants of the classical basically Nuclear Localization Signals (NLS) (aa 73–94 and 136–152) (ELM software (see Section Software and databases)) (Figure 1). The importance of the N-terminal region in the maintenance of TRIP12 nuclear localization was demonstrated in our recent study [7]. Within the nucleus, two studies noted an absence of TRIP12 in the nucleoli [7,15].

## 3. TRIP12 Protein Structure and Domains

TRIP12 is a large protein with an apparent molecular weight estimated between 189 and 227 kDa for the long forms (Table 1). It comprises several characterized protein domains: a catalytic HECT domain, two protein/protein interaction domains (WWE (Tryptophan-Tryptophan-Glutamate) and ARM (Armadillo repeats)), and a recently identified intrinsically disordered region (IDR) (Figure 1). These domains confer structure and functional roles to the protein. TRIP12 protein domains are conserved during evolution and are subject to post-translational modifications that likely modulate its structure and activity. Several studies reported HECT, ARM repeats, and WWE crystal structure of other proteins [16,17,18]. However, TRIP12 has yet to be recovered in crystal form to establish its complete 3D structure.

### 3.1. The HECT Domain

HECT domains (≈40 kDa) were first identified in the C-terminal part of the viral E6-AP protein; hence their name, Homologous to the E6-AP C-Terminus [2]. They are generally located in the C-terminal extremity of proteins and are constituted of two lobes: N-lobe domains (N-terminal end) and a C-lobe (C-terminal end) linked by a hinge domain [19,20]. The N-lobe is responsible for the interaction with the ubiquitin-conjugating enzyme E2 during the ubiquitination reaction (detailed in Section 4.1). The C-lobe carries the characteristic catalytic cysteine residue [21,22], which is highly conserved between species and essential to the ubiquitination reaction [2]. The TRIP12 HECT domain is located at residues 1552–1592, is structured in N and C-lobes, and contains a cysteine in position 1959 for which the replacement by an alanine residue abolishes TRIP12 ubiquitin ligase activity (Figure 1) [2,23]. Mechanistically, the 28 known HECT E3 ubiquitin ligases form an intermediate complex with ubiquitin before linking it to a lysine substrate via a thioester bond [20].

By ensuring the ubiquitin ligase catalytic activity, the HECT domain allows the implication of TRIP12 in different modes of protein degradation and signaling pathways (detailed in Section 4 and Section 5).

### 3.2. The WWE Domain

A WWE domain is a globular region of ≈80 amino acids characterized by an enrichment with hydrophobic and aromatic residues in its two extremities and the conservation of two tryptophan residues (W) and one glutamate (E) [24]. The C- and N-terminal parts of WWE domains form β-sheets, while the central part is an α-helix. WWE domains are required for protein–protein interaction. They are essential for two families of proteins: E3 ubiquitin ligases (HECT, RING, and RING-H2 types) and in a sub-family of poly (ADP-ribose) polymerases (PARP). PARPs add multiple ADP-ribose moieties to protein targets (parylation) and participate in various cellular functions such as DNA repair and chromatin dynamics. The WWE domains were demonstrated as poly(ADP-ribose) (PAR) binding domains [25]. The WWE domain of TRIP12 is located between the residues 749 and 836 (InterProScan) [26] and is required for the interaction of TRIP12 with APP-BP1 (Amyloid Protein-Binding Protein 1) [4], PTF1a (Pancreas Transcription Factor 1a) [6], and parylated PARP1 (Poly-(ADP ribose) polymerase 1) [27] (Figure 1).

### 3.3. The Armadillo-Repeats Domain

Armadillo (ARM) repeat-containing proteins are characterized by the presence of imperfect tandem repeats of 42 hydrophobic amino acids. An ARM repeat is composed of three α-helices (H1, H2, and H3). Successive ARM repeats are able to assemble themselves to form a super-helix also named a solenoid. As such, ARM-repeat motifs are strongly implicated in protein/protein interactions. ARM repeat-containing proteins are involved in pleiotropic functions (cell/cell adhesion, cell signaling, cytoskeleton organization, nuclear trafficking, and ubiquitination) [28]. ARM repeats were first identified in the TRIP12 orthologues KAKTUS (*Arabidopsis thaliana*) and UFD4 (Ubiquitin Fusion Degradation 4) (*Saccharomyces cerevisiae*) (detailed in Section 4.3) [29,30]. The InterProScan database shows that ARM repeats are also present in the human TRIP12 protein. Slightly different ARM-repeat boundaries are found in this database; we arbitrarily chose to represent the smallest and the most quoted one between residues 437 and 715 (Figure 1). Functional studies demonstrated the role of ARM repeats of UFD4 in the recognition of the ubiquitin degradation signal (detailed in Section 4.3) of the substrates in *Saccharomyces cerevisiae* [30].

### 3.4. The IDR Domain

Intrinsically disordered regions (IDR) are common in the human proteome, whereby an estimated 44% of the proteome carries an IDR. These regions can be functional but interestingly lack a defined 3D conformation in physiological conditions [31]. The interaction of an IDR-containing protein with a partner can stabilize and/or fold the disordered region into a stable secondary structure. IDRs are sequences enriched in basic, positively charged, and hydrophilic amino acids such as lysine (K), arginine (R), serine (S), and histidine (H). IDR-containing proteins are mostly involved in protein, DNA, and RNA binding/interaction [32,33,34]. Moreover, the presence of N-terminal IDRs is linked to DNA interaction [33,34,35]. We previously showed that TRIP12 interacts with chromatin through an IDR, which corresponds to the N-terminal quarter of the protein (aa 1–440) (Figure 1), and that the deletion of this domain leads to a loss of TRIP12 interaction with chromatin [7]. E3 ubiquitin ligases containing IDR structures are infrequent. Indeed, the IDEAL database (see Section Software and databases) references only 25 E3 ubiquitin ligases containing IDR among the ≈600 E3 ubiquitin ligases. According to the IDEAL (Intrinsically Disordered proteins with Extensive Annotations and Literature) database, TRIP12 IDR displays identity with well-characterized IDR-containing proteins such as MED1 (Mediator Complex Subunit 1, 25% of identity), AF9 (ALL1-Fused gene from chromosome 9, 23% of identity), and AF4/FMR2 (ALL1-Fused gene from chromosome 4/Fragile X Mental Retardation, 22% of identity). Numerous IDR-containing proteins are involved in transcription regulation [36], and these three proteins promote RNA polymerase II catalytic activity by being a coactivator (i.e., MED1) or by being part of the Super Elongation Complex (i.e., AF9, AF4/FMR2) [37,38]. Taken together with the discovery of TRIP12 chromatin interaction, TRIP12 may act as a new transcriptional regulator. Liu et al. described an APC (Adenomatous polyposis coli) domain, which is known to be positively charged and involved in microtubules interaction, within TRIP12 IDR [39]. However, the function of this APC domain was not further explored. It is also now well documented that IDRs can confer to proteins the particular biophysical property of the liquid-liquid phase separation (LLPS) by which two liquids form two different phases such as oil droplets in water [40,41]. This biophysical process was linked to genome organization and expression [42].

### 3.5. Conservation of TRIP12 Protein Domains during Evolution

TRIP12 orthologues were differently named depending on the species. For example, the TRIP12 orthologue is CTRIP (Circadian TRIP) in *Drosophila melanogaster*, KAKTUS (KAK also known as UPL3), in *Arabidopsis thaliana* (UniProt (see Section Software and databases)) and Ubiquitin Fusion Degradation protein 4 (UFD4) in *Saccharomyces cerevisiae* (UniProt). The size of TRIP12 orthologues also varies per species. The largest form is found in *Drosophila melanogaster* (3140 aa), while the smallest sequence is in *Saccharomyces cerevisiae* (1483 aa) (Figure 3). The TRIP12 amino-acid sequence is highly conserved in vertebrates, demonstrating 82% identity for the zebrafish to 97.7% for the mouse when compared to the human c isoform. This percentage of identity with the human sequence decreases in invertebrates, 38% for *Drosophila melanogaster*, 31.4% for *Arabidopsis thaliana*, 28% for *Schmidtea mediterranea*, and 27.2% for *Saccharomyces cerevisiae*. HECT and ARM-repeats domains are found in all species, whereas the WWE domain is found among all vertebrates and *Schmidtea mediterranea* but not in *Saccharomyces cerevisiae* and *Drosophila melanogaster*. The HECT domain, which is required for polyubiquitination, is present in less evolved species, underlying the importance of E3 ubiquitin ligase activity of TRIP12 during evolution. IDRs are fast evolving regions. Their evolution rate is higher than of structured domains due to the lack of 3D structure [31]. Highly conserved IDRs are often found in proteins involved in transcription regulation and DNA binding [31]. An N-terminal IDR is present in all the TRIP12 orthologues mentioned above. According to IUPred (see Section Software and databases), the TRIP12 IDR length varies from 1,042 aa in *Drosophila melanogaster* CTRIP to 117 aa in *Schmidtea mediterranea*. Vertebrate TRIP12 orthologues showed an IDR average size of 440 residues. Taken together, the high conservation of TRIP12 domains during evolution assumes essential functions for this protein in living organisms.

### 3.6. Post-Translational Modifications of TRIP12

Similar to many other proteins, TRIP12 is post-translationally modified. The PhosphoSitePlus^®^ database (see Section Software and databases) provides an exhaustive list of the post-translationally modified residues identified by mass spectrometry and other approaches (Figure 4). Among the modifications, the most abundant are the phosphorylation and the ubiquitination. TRIP12 can also be acetylated and mono-methylated. TRIP12 IDR is subject to abundant post-translational modifications, since it is a common feature of IDR domains [31], which is in contrast to the ARM repeats-domain that is poorly modified. Only few phosphorylation sites were attributed to specific kinases such as the serine S424, S1078, and S1577 to the kinases CDK1 (Cyclin Dependent Kinase 1), CHK1 (Checkpoint Kinase 1), and ATM/ATR (Ataxia-Telangiectasia-Mutated/Ataxia Telangiectasia and Rad3-related), respectively [44,45,46] (Figure 1). Despite two independent studies that reported the ubiquitination of TRIP12 [47,48] and numerous identified ubiquitinated residues, the ubiquitin ligases responsible for TRIP12 ubiquitination are unknown. Moreover, we cannot exclude that TRIP12 may exhibit functions of self-ubiquitination. It was demonstrated that the Ubiquitin-Specific Peptidase USP7 deubiquitinates and stabilizes the TRIP12 protein [47,48]. Despite numerous studies dedicated to TRIP12, none of the post-translational modifications are shown to function in the regulation of catalytic activity, interacting capacity, nor in the cellular localization of this E3 ubiquitin ligase.

## 4. Functions of TRIP12 in Ubiquitin-Mediated Proteolysis

A detailed description of ubiquitination process, different ubiquitin ligase families, and functions are excellently reviewed elsewhere [20,49,50,51]. In this section, we will provide an overview of the essential information on TRIP12 actions in ubiquitin-dependent proteolytic pathways.

### 4.1. Ubiquitination

Ubiquitination is an essential ATP-requiring process leading to protein modification by the covalent conjugation of ubiquitin, which is a highly conserved 76 aa polypeptide (≈8.5 kDa) [49]. Although best known for targeting proteins to degradation by the 26S proteasome, ubiquitination is potentially linked to all cellular processes. Canonical ubiquitin conjugation involves the attachment of the C-terminal glycine 76 (G76) of ubiquitin (Ub) to the ε-amino group of a lysine on the protein substrate. Substrates can be either modified by a single ubiquitin on one (monoubiquitination) or more sites (multi-mono-ubiquitination) or by chains of ubiquitin [51]. In the case of poly (or multi) Ub chains, the C-terminal G76 of one Ub molecule is linked to the ε-NH_2_ group of one of the seven lysines (K6, K11, K27, K29, K33, K48, or K63) of the preceding ubiquitin. The poly-Ub chains can have different lengths and topologies resulting in different consequences on the fate of the modified protein substrate. For instance, K48-linked polyubiquitination primes proteins for proteolytic destruction by the proteasome, whereas non-K48 linkages target proteins for various cellular functions and responses [50,51].

Ubiquitination is ensured by a three step-concerted action of three enzymes. An Ub-activating enzyme (E1) catalyzes the formation of a thiol-ester linkage between the C-terminal lysine G76 of Ub and the active cysteine (C) residue of E1; the Ub molecule is transferred to the cysteine of the Ub-conjugating enzyme (E2). E3 ligases are classified into three major classes, the RING (Really Interesting New Gene), the RBR (RING-between RING-RING), and the HECT ligases. For HECT and RBR E3 ligases, the transfer of Ub to the substrate lysine-amino group involves a thioester intermediate with a conserved cysteine in the catalytic site of the HECT domain or the Rcat domain, respectively [20]. The RING E3 ligases catalyze the direct transfer of ubiquitin from a thioester-linked E2-ubiquitin conjugate to a substrate [20]. In some cases, the cooperation of an additional E4 ubiquitin ligase, which may encode for independent ligase activity or enhance the processivity of the E3, is required. To act as catalysts, E3s recruit their substrates for degradation by recognizing a specific portion in the protein sequence or degron. Ubiquitin conjugation can be reversed by deubiquitinating enzymes (DUBs) [52].

### 4.2. Known TRIP12 Substrates

Based on the literature, TRIP12 regulates the stability and promotes the proteolysis of nine protein substrates: ASXL1 (Additional Sex Combs Like 1) [53], USP7 (Ubiquitin-Specific Peptidase 7) [39], PTF1a (Pancreas Transcription Factor 1a) [6], SOX6 (SRY-Box Transcription Factor 6) [12], RNF168 (Ring Finger Protein 168) [14], BAF57 (BRG1-Associated Factor 57) also known as SMARCE1 (SWI/SNF Related, Matrix Associated, Actin-Dependent Regulator Of Chromatin, Subfamily E, Member 1) [54], P14/ARF (P14/Alternate Reading Frame) also known as CDKN2A (Cyclin Dependent Kinase Inhibitor 2A), for which TRIP12 was named ULF (Ubiquitin Ligase for ARF) [15], APP-BP1 (Amyloid Protein-Binding Protein 1) also known as NAE1 (NEDD8-Activating Enzyme E1 Subunit 1) [4] and PARP1 (Poly-(ADP ribose) polymerase 1) [27]. These proteins exert important functions in biological and/or pathological processes that will be outlined in Section 5.

The TRIP12 HECT domain (detailed in Section 3.4) is essential for the degradation of these proteins, but the role of TRIP12 as direct interacting E3 ligase is still questionable for ASXL1 [53] and RNF168 [14] for which experimental approaches that prove ubiquitination by TRIP12 are lacking. TRIP12 polyubiquitinates its substrates with the exception of APP-BP1, which is monoubiquitinated [4]. APP-BP1 polyubiquitination requires an additional E4 activity [4]. TRIP12 mediated-K48-linked polyubiquitination was demonstrated only for USP7 [39] and PTF1a [6]. While the interaction between the E3 ligase and its substrates was validated, the interaction domains of TRIP12 have been mapped only in few cases. Although these studies used different isoforms of TRIP12, they all emphasize the role of consensus protein/protein interaction domains of TRIP12. A region (aa 611–1259 of isoform c) that includes the WWE domain but not ARM repeats is responsible for the interaction with APP-BP1 [4]. A sequence of 577 aa (aa 234–810 of isoform a) preceding the WWE domains interacts with USP7 [39]. In addition, TRIP12 interacts with parylated PARP1 via its WWE (aa 797–911 of isoform a) domain. TRIP12 is freshly considered as an identified PAR-targeted ubiquitin ligase (PTUbL) controlling PARP1 turnover [27]. Finally, we demonstrated that PTF1a binds a region containing ARM repeats and the WWE domain (aa 446–1552 of isoform c) [6].

Interacting domains or ubiquitinated residues were identified for certain proteins regulated by TRIP12. A sequence of 80 aa within and flanking the central coiled-coil region of BAF57 was found to be critical for BAF57 stabilization and to contain lysine residues essential for ubiquitination [55]. The N-terminal part of Tumor necrosis factor-receptor associated factor (TRAF), also known as the Meprin And TRAF-C Homology (MATH) domain of USP7, is responsible for its interaction with TRIP12 [39]. We found that the lysine 312 of PTF1a is essential for its TRIP12-mediated degradation [6]. It was previously reported that the tumor suppressor ARF is a natural lysine-less protein that undergoes N-terminal polyubiquitination [56].

Intriguingly, TRIP12 targets several proteins that are unbound to their physiological binding partners. This is the case for BAF57, which is a subunit of the SWI/SNF (SWItch/Sucrose Non-Fermentable) chromatin remodeling complex (detailed in Section 5.4.) [54]. Most if not all SWI/SNF subunits are assembled into a complex for which maintenance of the stoichiometry is essential and no free subunits exist in the cell; otherwise, they are degraded. The stabilization and protection of BAF57 from proteasome-mediated degradation operates via protein–protein interaction with the BAF155 subunit [57]. TRIP12 was demonstrated to compete with BAF155 for BAF57 and to degrade only an unbound form of BAF57 [55]. The same type of interaction was found for the ubiquitination of the neddylation E1 enzyme subunit APP-BP1 [4]. In fact, TRIP12 does not interact with APP-BP1 in the heterodimeric form bound to Uba3 (Ubiquitin-Like Modifier-Activating Enzyme 3), which is the specific active E1 of the NEDD8 pathway. A similar scheme is described for ARF, which is stabilized and protected from TRIP12 ubiquitination when sequestered by nucleophosmin and TRADD (NFR1-Associated Death Domain Protein) in the nucleolus (detailed in Section 5.2.) [15,58]. Indirect evidence suggests a similar mode of action for the degradation of RNF168 [14]. Indeed, the destabilization of RNF168 induced by a silencing of its binding partner HERC2 (HECT and RLD Domain Containing E3 Ubiquitin Protein Ligase 2) could be mitigated by a depletion of TRIP12. The TRIP12-ubiquitinated lysine 312 of PTF1a is located near the domain interacting with RBP (mammalian suppressor of Hairless) essential for the transcriptional activity of PTF1a. One may hypothesize that free PTF1a could be ubiquitinated, while binding to RBP could prevent TRIP12-mediated PTF1a degradation. Therefore, TRIP12 appears as a regulator of the balance between bound and unbound forms of its protein substrates to maintain them in their complexed forms.

### 4.3. Role of TRIP12 in Ubiquitin Fusion Degradation and N-Degron Pathways

TRIP12 is sequelogous to the UFD4 HECT E3 ligase, which catalyzes the ubiquitination of Ubiquitin Fusion Degradation (UFD) substrates in yeast [59]. These substrates are artificial fusion proteins consisting of an N-terminal uncleavable ubiquitin moiety that acts as a degron. The UFD pathway consists in the poly-ubiquitination of the uncleavable N-terminal ubiquitin. TRIP12 was shown to mediate the degradation of the aberrant form of ubiquitin named UBB^+1^ [60]. UBB^+1^ is a physiological human UFD substrate resulting from a dinucleotide deletion in the mRNA of the ubiquitin B gene. UBB^+1^ accumulates in the brain in neurodegenerative disorders [61]. The HECT E3 HUWE1 (HECT, UBA, and WWE domain containing E3 ubiquitin protein ligase 1) was reported to cooperate with TRIP12 for the degradation of UBB^+1^ [62]. Although the conservation of the UFD pathway from yeast to human attests its importance for proteolysis in cells, except for UBB^+1^, no other native substrates have been identified. Most studies use artificial substrates for specific applications. In particular, enhanced immune response after UFD ubiquitination of antigens opens up interesting perspectives in the development of tumor vaccines [63].

The N-degron pathway (formerly N-end rule pathway) is another ubiquitin-dependent proteolytic pathway that regulates the half-life of proteins according to N-terminal degradation signals or N-degrons. UBR1, UBR2, UBR4, and UBR5 (ubiquitin protein ligase E3 component N-recognin) are the four E3s (or N-recognins) encoded by the mammalian genomes that can recognize Arg/N-degrons [64]. Interestingly, a study demonstrated that UBR1 and UFD4 E3s interact and cooperate to enhance substrate ubiquitination in both pathways in yeast [65]. This mutually cooperative physical interaction between E3s operating in parallel pathways are likely to be relevant to all eukaryotes and might apply to TRIP12 and UBR5 control of RNF168 accumulation [14].

## 5. The Physio-Pathological Roles of TRIP12

Proteomic approaches have allowed for the identification of several TRIP12 binding partners (Figure 5 and Table 2). However, the molecular function of only a restricted number of these interactions is known. Thus far, TRIP12 was implicated in the regulation of several important molecular pathways such as cell cycle progression, DNA damage repair (DDR), chromatin remodeling, and cell differentiation via a small number of identified protein interactors, which leaves wide open the field of investigation of molecular processes implicating TRIP12.

### 5.1. TRIP12 Protein Interactors

According to BioGrid resources (see Section Software and databases) and the literature, 76 unique proteins interact with human TRIP12 (Table 2). They were identified by different experimental approaches using either these proteins (*n* = 58), either TRIP12 (*n* = 7) or both as bait (*n* = 11). Among them, eight were further classified as TRIP12-ubiquitinated substrates (detailed in Section 4.2). A STRING protein–protein network and functional enrichment analyses (see Section Software and databases) reveals that 76% (58 out of 76) are implicated in cellular macromolecule metabolic process (red circles), which corresponds to proteins involved in the adding of low molecular mass protein units (i.e., ubiquitin) (Figure 5). In parallel, 42% (32 out of 76) are involved in the positive regulation of nucleobase-containing compound metabolism (purple circles), which corresponds to cellular processes that activate chemical reactions/pathways involving nucleic acid-related compounds. Moreover, we can clearly distinguish two protein-protein functional clusters including proteins involved in transcriptional regulation (i.e., PTF1a, EZH2, SUZ12, etc.) and in the ubiquitination/deubiquitination system (i.e., USP7, VHL, USP11, etc.). However, it is important to keep in mind that all the interactions of the TRIP12 with protein partners might depend on the type of cells, the stage of differentiation, or the cell cycle phase. Altogether, these analyses position TRIP12, via its interacting partners, as a potential central regulator of protein stability and genome expression.

### 5.2. Roles in Cell Cycle Progression

The P53 dependent-activating P14/ARF (P14/Alternative Reading Frame also known as CDKN2A (Cyclin-Dependent Kinase Inhibitor 2A)) pathway plays a major activating role in cell cycle progression. Chen et al. identified TRIP12 as the E3 ubiquitin ligase responsible for the degradation of ARF and renamed it ULF for Ubiquitin ligase for ARF [96]. TRIP12 decreases the stability of ARF after polyubiquitination and degradation by the proteasome. This leads to a proteasome-mediated destabilization of P53 by the E3 ubiquitin ligase MDM2 (Mouse Double Minute 2 homolog) and a cell cycle progression (Figure 6, left panel). The authors further showed the relationship between TRIP12 and two positive regulators of ARF: the nucleophosmin (NPM), a major component of nucleoli, and the transcription factor c-MYC. They reported that NPM sequestrates ARF in the nucleoli, isolating it from the ubiquitinating action of TRIP12 present only in the nucleoplasm (Figure 6, left panel). Then, they demonstrated that c-MYC enhances the ARF half-life by destabilizing the TRIP12/ARF nucleoplasmic complex (Figure 6, right panel). Therefore, the interaction of ARF with MDM2 prevents the ubiquitination-mediated degradation of P53, which in turn, induces a cell cycle arrest and apoptosis [114]. Later, the nucleostemin (also known as Guanine nucleotide-binding protein-like 3) was demonstrated to stabilize the NPM/ARF complex in the nucleoli and to destabilize the TRIP12/ARF complex in the nucleoplasm to inhibit ARF degradation by TRIP12 ubiquitination [75]. The translocation of TRADD (TNF-R Associated Death Domain Protein) from the cytoplasm to the nucleus was also shown to protect ARF from degradation by inhibiting its interaction with TRIP12 [58]. The participation of TRIP12 in cell cycle progression was assessed in non-cancerous cells subject to an oncogenic stress provoked by an overexpression of c-MYC [115]. Paradoxical properties of c-MYC overexpression have been reported in oncogenic processes [116]. Indeed, a mild expression of c-MYC stimulates cell proliferation, whereas a strong overexpression inhibits cell proliferation by activating the ARF/P53 response. In both cases, ARF mRNA levels were induced. However, the authors demonstrate that only a high concentration of c-MYC is able to inhibit the degradation of ARF mediated by TRIP12, which in turn leads to the activation of the ARF/P53 pathway and cell cycle arrest. Moreover, Kajiro et al. generated TRIP12 knock-in mutant mice by homozygous mutation in *TRIP12* exon 33 (*TRIP12^mt/mt^*) disrupting the ubiquitin ligase activity [5]. This resulted in embryonic lethality at the embryonic stage E11.5. *TRIP12^mt/mt^* ES cells exhibited a significant decrease of growth rate caused by the abnormal induction of the cyclin-dependent kinase inhibitor P16 and ARF associated to a P53 protein increased stability. *TRIP12^mt/mt^* ES cells also display an abnormal accumulation in G_2_/M and sub-G_1_ phases. These data support that TRIP12 functions in the cell cycle control in vivo.

TRIP12 is tightly implicated in the control of cell cycle progression. Until very recently, little was known on cell cycle regulation of TRIP12 expression. We showed that the presence of TRIP12 in the nucleus strongly depends on the phase of the cell cycle. The TRIP12 protein appears in the nucleus in the late S phase and endures during the G_2_ phase [7]. We found that while tightly bound to the chromosomes during mitosis, TRIP12 is rapidly degraded in the early G_1_ phase and disappears in the nucleus in the late G_1_ and early S phase. We further demonstrated that TRIP12 plays a role in phase S by controlling DNA replication timing. Via its chromatin-interacting IDR, TRIP12 coordinates mitotic entry and progression [7]. The important function of TRIP12 during mitosis was reinforced by mass spectrometry-based approaches showing that TRIP12 is specifically retained to mitotic chromatin [117]. With regard to variations to TRIP12 expression during the cell cycle, it would be important to reconsider TRIP12 molecular functions according to its presence or absence from the nucleus.

### 5.3. Roles in DNA Damage Response

#### 5.3.1. Via the ARF/P53 Signaling Pathway

Oncogenic stimuli trigger the activation of checkpoints that delay or prevent tumorigenesis. The DNA damage repair (DDR) pathway and ARF protein are two major checkpoints that both act by the protein P53. Initially studied independently, the interconnection between these two checkpoints in tumorigenesis is gaining notoriety. For example, Velimezi et al. discovered that the key DDR kinase ATM negatively regulates the abundance of ARF with the implication of TRIP12 [118]. In response to DNA damage, the kinase ATM activates the protein phosphatase 1 (PP1), which in turn, dephosphorylates the nucleolar protein NPM (Figure 7). This dephosphorylation liberates ARF from NPM, rendering it susceptible to degradation via the action of TRIP12 in the nucleoplasm. Chen et al. further reported the contribution of TRIP12 in ARF degradation in response to DNA damage induced by doxorubicin [115]. The data showed that after DNA damage, the degradation of ARF is strongly attenuated in TRIP12 knockout mouse embryonic fibroblasts (MEFs). Another study identified the serine S1577 of TRIP12 (Figure 1) as phosphorylated by the kinases ATM/ATR, which suggests that post-translation modifications are possibly critical for a proper implication of TRIP12 in the DDR network [45].

#### 5.3.2. Via the Control of Histone Ubiquitination Spreading by RNF168

Histone ubiquitination plays a crucial role in the recruitment of proteins responsible for the detection and the repair of DNA damage. The E3 ubiquitin ligase RNF8 (Ring Finger Protein 8) initiates the ubiquitination of histones H2A and H2AX at the damaged sites (Figure 7) [119]. This priming of ubiquitination reaction is amplified by another E3 ubiquitin ligase, RNF168. The produced polyubiquitination (K63 chain) chain acts as a platform for the interaction of repair signaling factors, such as 53BP1 and the complex RAP80/BRCA1 (Receptor-Associated Protein 80/Breast Cancer 1) [122]. If not properly controlled, RNF168 over-amplifies the ubiquitination signal beyond the DNA breakage on neighboring chromatin, which can have dreadful consequences, notably by sequestrating the genome keepers 53BP1 (P53 Binding Protein 1) and BRCA1. The retention of 53BP1 and BRCA1 at a non-damaged DNA site prevents their recruitment at damaged DNA sites and subsequent repair. It was shown that TRIP12 and UBR5 are two proteins that prevent this over-amplification after the DNA damage induction of double-strand break (DSB) [14]. In physiological conditions, TRIP12 and UBR5 control the quantity of RNF168 required to restrain histone ubiquitination around DNA lesions (Figure 7). A depletion of TRIP12 and UBR5 provokes an accumulation of RNF168 to supra-physiological levels that amplifies histone ubiquitination beyond the DSB sites. Moreover, TRIP12 was demonstrated as the E3 ubiquitin ligase implicated in the stability of RNF168.

#### 5.3.3. Via the Deubiquitinase USP7

It is accepted that the deubiquitinase USP7 plays a major role in the response of DNA damage by stabilizing the expression of key proteins such as P53 and RNF168 [120,121]. TRIP12 interacts with USP7 and polyubiquitinates (K48 chain) it to induce its degradation by the proteasome (Figure 7) [39]. Consequently, the TRIP12-mediated degradation of USP7 affects the stability of 53BP1 and CHK1 proteins, which are two important regulators of DDR regulated by USP7. Importantly, as mentioned in Section 3.6, the TRIP12 protein is also stabilized by USP7, suggesting a relevant interconnected feedback between these proteins. Taken together, these studies show that TRIP12 and USP7 mutually control their expression level. In addition, they both regulate the expression level of RNF168, which is a key factor in DSB repair. This indicates that a fine-tuning of these three enzymes is certainly required to ensure a proper managing of DNA damage repair via RNF168 quantity control.

### 5.4. The Roles of TRIP12 in Chromatin Remodeling

#### 5.4.1. Via the Control of SWI/SNF Complex Integrity

The level of chromatin compaction participates in highly dynamic processes such as transcription, DNA repair, and recombination. When the chromatin is compacted, genes are globally transcriptionally inactive due, in part, to a low accessibility of the transcriptional machinery to promoters [123]. The modification of chromatin compaction and consequently the modulation of transcriptional activity is partially controlled by the SWI/SNF complex [124]. It is a multi-protein complex constituted by a dozen sub-units. The sub-units BRG1 (hSWI/SNF Brahma-related Gene) and BRM-1 (BRachyury Modifier 1) ensure the ATPase catalytic activity. The function of the SWI/SNF complex is to remodel nucleosome positioning by sliding, inserting, or ejecting histone octamers into DNA in an ATPase-dependent manner. As such, it controls numerous cellular processes [125]. To reach an optimal efficiency, the SWI/SNF complex requires the presence of BAF proteins (BRG-1 associated factors), which interact with BRG-1. Therefore, the regulation of expression of BAF proteins by proteasome-dependent degradation is crucial. It was demonstrated that TRIP12 is the main E3 ubiquitin ligase that polyubiquitinates the free cellular pool of BAF57 (also known as SMARCE1 (SWI/SNF-related matrix-associated actin-dependent regulator of chromatin subfamily E member)) by a mechanism that we described in Section 4.2. [54]. Therefore, TRIP12 appears to be involved in a protein quality control system of the SWI/SNF complex. The TRIP12 mediated-degradation of BAF57 could affect the loading of SWI/SNF on its target genes and consequently disturb gene expression.

#### 5.4.2. Via the Maintenance of Silenced Genes by the Polycomb Complex

Epigenetic modifications of DNA bases play a crucial role in the control of gene transcription. Most of these modifications were first identified in prokaryotes and then in higher eukaryotes. The methylation of the nitrogen atom in position 6 of an adenine (6mA) is the most frequent modification in prokaryotes. This modification was recently described in eukaryotes and particularly in vertebrates [126]. In mammals, the methyltransferase METTL44 (Methyltransferase Like 4) and the dioxygenase ALKBH4 (ALKB Homolog 4, Lysine Demethylase) are responsible for the deposit and the suppression of the 6mA on DNA, respectively [53]. In mammals, the 6mA is associated with epigenetic repressive marks such as the mono-ubiquitination on the K119 of histone H2A (K119 Ub-H2A) and the trimethylation on K27 of histone H3 (K27me3-H3). These repressive marks are used for silencing promoters during the stem cells differentiation and embryonic development. They are mainly added on the chromatin by the Polycomb Repressive Complexes (PRC1 and PRC2) [127,128]. These two marks are reversible. The K119 Ub-H2A is removed by the deubiquitinase complex PR-DUB (Polycomb Repressive-Deubiquitinase complex) formed by the proteins ASXL1 (Additional Sex Combs like 1) and BAP1 (BRAC1 Associated Protein 1) [129]. The BAP1 protein bears the deubiquitinase activity, and it is stabilized by ASXL1 [130]. Mass spectrometry based-proteomic analyzes identified the protein-protein interaction between TRIP12 and major components of PRCs such as SUZ12 (Polycomb protein SUZ12), EZH2 (Histone-lysine N-methyltransferase) [84] and EED (Embryonic Ectoderm Development protein) [80] but with no mechanistic findings. A recent study characterized the role of 6mA deposit and the implication of the E3 ubiquitin ligase TRIP12 in the maintenance of the repressive marks added by the Polycomb complex [53]. The addition of 6mA on PRC1/PRC2-inactivated genes leads to the recruitment of the PR-DUB complex. This recruitment on the chromatin leads to the degradation of the complex by TRIP12-mediated polyubiquitination of ASXL1 (discussed in Section 4.2.) and preserves the level of K119 Ub-H2A. A homozygous deletion of the *METTL4* gene in a murine model induces a decrease of 6mA deposit and provokes a craniofacial dysmorphism in a subset of pups. Interestingly, de novo nonsense mutations of the *ASXL1* gene were identified in patients with Bohring–Opitz syndrome who display facial dysmorphism [131]. In humans, mutations of the *TRIP12* gene are also associated to craniofacial dysmorphism (detailed in Section 6.1). Therefore, we can speculate that an inactivation of TRIP12 could provoke an accumulation of ASXL1 and a stabilization of the PR-DUB complex that would subsequently alter the expression of genes implicated in facial development.

### 5.5. Roles in Cell Differentiation

Different studies have proven the strong involvement of TRIP12 in the cell differentiation. Differentiation is the process by which dividing cells change their functional or phenotypical type. It involves the coordinated regulation of genes by a multitude of cellular pathways. The first study showing the participation of TRIP12 in stem cells and regeneration mechanisms was performed in the freshwater planarian *Schmidtea mediterranea* [132]. *Schmidtea mediterranea* is a model for regeneration, stem cells, and the development of tissues. This worm is able to replace cell loss during normal cellular turnover or tissue loss through the regenerative capacity of adult stem cells called neoblasts. The authors demonstrated that 11 genes of HECT E3 ubiquitin ligases were expressed in neoblasts. Using RNA interference (RNAi), the authors showed that the TRIP12 orthologue also known as “smed-TRIP12” is essential for the establishment and the maintenance of posterior tissues.

TRIP12 displays another important function in the control in muscle fiber homeostasis. An et al. showed that the transcription factor SOX6 (SRY-Box Transcription Factor 6) is a target of TRIP12 in mammals. SOX6 is involved in muscle fiber homeostasis by the transcriptional repression of “slow switch muscle fiber” genes in fetal mice [133]. They showed that a TRIP12 knockdown in myotubes-derived cell lines leads to an increase of SOX6 protein levels and results in a dysregulation of muscle fiber homeostasis. TRIP12 binds to SOX6 protein, inducing its polyubiquitination and its degradation by the proteasome [12].

TRIP12 is also implicated in the homeostasis of pancreatic epithelial cells. We demonstrated that TRIP12 polyubiquitinates and provokes the degradation of the transcription factor PTF1a (Pancreas Transcription Factor 1a) by the proteasome [6]. PTF1a is a transcription factor that controls the expression of pancreatic acinar specific genes such as amylase and chymotrypsin genes. This factor is essential in pancreatic development and pancreatic cell homeostasis by maintaining the acinar phenotype. The expression of PTF1a is lost during acinar to ductal cells transdifferentiation [133,134]. PTF1a is also required for the specification of inhibitory neurons in various regions of the developing central nervous system including the spinal cord, brainstem, cerebellum, and retina [135,136,137]. Thus, it is possible that TRIP12 regulates neuronal specification via PTF1a degradation.

## 6. TRIP12 in Pathologies

### 6.1. Alterations of TRIP12 Gene in Autism and Intellectual Disability

*TRIP12* (OMIM 604506 (Online Mendelian Inheritance in Man) (see Section Software and databases)) is a neurodevelopmental disorder (NDD)-associated gene. Several publications allowed its identification as a primary gene for Intellectual Disability (ID), which is the most common developmental disorder affecting 1–3% of the world’s population. Most IDs are monogenic with 1,396 genes causing ID [138]. The *TRIP12* gene is referenced as a causative gene associated with Clark–Baraitser syndrome (CLABARS; OMIM617752), formerly autosomal-dominant mental retardation-49 (MRD49) in the OMIM database. Although originally thought to be an X-linked disorder [139], Clark-Baraitser syndrome was shown to have an autosomal dominant mode of inheritance [140]. In addition to mental retardation, craniofacial dysmorphism, behavioral psychiatric manifestations, and obesity are the main clinical features of the syndrome. CLABARS is caused by heterozygous mutation in the *TRIP12* gene.

The first indication of a putative role of the *TRIP12* gene in ID was the report of a large interstitial ≈5.6-Mb deletion on the long arm of chromosome 2 (2q36.2q36.3) in a 17-year-old girl. Twenty-four genes were located in the deleted fragment where *TRIP12* was disrupted. The phenotype included severe mental retardation, facial dysmorphism, and multiple renal cysts [141]. The identification of *TRIP12* among 10 new ID genes was later accomplished by meta-analysis of large-scale exome sequencing data of a combined cohort including 2104 patient-parent trios [142] where two different de novo heterozygous mutations in the *TRIP12* gene were identified in two unrelated patients.

Of importance, several de novo mutations (DNM) of the *TRIP12* gene were associated with a broader phenotype than ID in line with the overlap of genes and pathways involved in NDDs. Mutual comorbidities of ID and Autism Spectrum Disorder (ASD) also raised the question of whether *TRIP12* can be validated as a gene associated with ASD. Resequencing sixty-four ASD risk genes in two ASD cohorts newly identified mutations in the *TRIP12* gene, thus proposing that it could be a novel candidate gene for clinical follow-up [143,144]. Recent detailed clinical information characterizing the phenotype of 11 individuals with mild to moderate ID or learning disability finally demonstrated that *TRIP12* is a risk gene for ID with (8/11) or without ASD [10]. However, there is no correlation for the presence or absence of autistic features with the type of mutations in this study. Other typical clinical features of the *TRIP12*-associated spectrum such as unspecific craniofacial dysmorphism, speech delay, and seizures were reported in another cohort of nine pathogenic variants that further documented that *TRIP12* haploinsufficiency causes childhood-onset neurodevelopmental disorders [145]. Recent exome sequencing of Japanese trios and integrative analysis conducted by combining published DNM data identify *TRIP12* among 61 significant genes for ASD and suggest that it is a genuine disease-causative gene [146]. The *Trip12* gene is a high confidence gene for autism in the SFARI GENE database (see Section Software and databases).

Pathogenic *TRIP12* variants are now documented in gene variant databases related to human health LOVD (Leiden Open Variation Database (see Section Software and databases)) and ClinVar (Clinical Variation (see Section Software and databases)) and can be also found in ID/autism websites. According to the LOVD, most pathogenic variants (87%) have DNA changes in coding regions, 8% in spliced regions, 5% in multiple regions. These changes are substitutions (53%), deletions (29%), duplications (13%), and 5% of these variants have still unknown modifications. Consequences at the protein level include stop changes (32% of variants) and frame shifts (26%) generating premature termination, missense changes (18%), no protein production (8%), or unknown (16%). Selected variants are placed on the TRIP12 protein sequence according to the transcript variant 3, NM_004238.3 (Figure 8).

Analysis was performed to identify protein/protein interaction networks and signaling pathways associated with a set of autism and ID genes. These frequently implicate neuronal signaling and development, chromatin remodeling, transcriptional regulation, mRNA splicing, ion transport, cell cycle, and microtubule cytoskeleton. They converge on neuronal and cognitive functions such as the regulation of synaptic transmission, learning, and memory [146,147,148]. To date, only one pilot study in *Drosophila melanogaster* investigated the role of TRIP12 using inducible RNA interference. This study reported no significant change in behavioral assay after *TRIP12* knock-down, suggesting no role in the development of the nervous system [148]. Given the important function for protein degradation in the nervous system, the synaptic plasticity, and the maintenance of neuronal structures, one might speculate that TRIP12 regulates the synaptic proteome. However, physiological roles of TRIP12 suggest its potential contribution to other essential pathways that are deregulated in ID pathologies. Therefore, additional functional analysis using relevant models is necessary to conclude a causal association between *TRIP12* gene variants and these diseases.

### 6.2. Alterations of TRIP12 Gene, mRNA and Protein Expression, and Function in Cancers

#### 6.2.1. Alteration of *TRIP12* Gene in Cancers

The cBioPortal database (see Section Software and databases) provides the frequency of CNA (Copy Number Alteration) appearance in *TRIP12* gene in cancers [149,150]. It includes mutations, fusions, amplifications, deep deletions, and multiple alterations (Figure 9). On average, the *TRIP12* gene is altered in 1.84% of cancer patients (over 27,235 patients). This rate remains relatively weak compared to other genes, for instance, *TP53* (39.8%). *TRIP12* gene alteration frequency varies from 11.7% in endometrial carcinoma to 1.06% in glioblastoma patients. Up to now, 463 mutations have been located in the human *TRIP12* gene (Figure 10). These punctual alterations do not accumulate in hot spots but rather spread all along the protein sequence. Among them, 323 were classified as missense, 120 were classified as truncating mutation, and one arises from nucleotides insertion without an open reading frameshift. Additionally, 20 alterations result from a fusion between the *TRIP12* gene and other genes such as PID1 (PTB-containing, cubilin and LRP1-interacting protein) or KIF1A (Kinesin-like protein). However, a nonsense mutation is retrieved 10 times at the residue S1195 generating a truncated protein: six times in colon cancer, two times in uterine cancer, and once in lung and breast cancer. The main issue regarding *TRIP12* gene alterations and in particular point mutations is whether these mutations play a causal role in cancer initiation or progression (“driver” mutations) or if they only randomly appear during cancer establishment as a consequence of, for instance, DNA repair failure (“passenger” mutations).

Only six publications have reported alterations in the *TRIP12* gene in cancers. Frameshift mutations of genes containing mononucleotide repeats are features of gastric and colorectal carcinomas with microsatellite instability (MSI). The *TRIP12* gene contains repetitions of seven adenines in exon 3 and seven thymidines in exon 24. In a cohort of 27 gastric and 32 colorectal cancer samples, Yoo et al. identified an adenine insertion in the exon 3-adenine repetition in one case of gastric cancer and a thymidine deletion in the exon 24-thymidine repetition in one case of colorectal cancer [151]. These frameshift mutations predict the formation of a truncated TRIP12 protein with no catalytic activity. Unfortunately, the authors did not investigate the functional consequences of this insertion. Another study aiming to identify the genetic loci with an elevated risk of breast cancer related to menopausal hormone therapy (MHT) in post-menopausal women showed a strong linkage between a Single Nucleotide Polymorphism (SNP) located in the exon 1-2 intronic region of the *TRIP12* gene and MHT [152]. However, they were not successful in replicating *TRIP12* SNP × MHT interaction in an independent cohort. Corresponding analyses of public data of global RNA expression were obtained from 12 lung adenocarcinomas and six controls to identify mutated genes with high-frequency-risk expression [153]. They identified the *TRIP12* gene as the most frequently mutated (missense mutation on exon 16) (nine cases out of 12). A recent study reported somatic mutations of the *TRIP12* gene in three out 20 cases of HPV-positive anal squamous cell carcinoma (HPV-ASCC) [154]. A missense mutation, a frameshift internal deletion, and a nonsense mutation were identified on exons 3, 37, and 33, respectively. However, the functional consequences of these mutations were not further investigated. A RNA-seq analysis of 139 patients with pediatric Acute Myeloid Leukemia (AML) revealed the existence in one patient of a new gene rearrangement leading to the fusion between TRIP12 and NPM (nucleophosmin) proteins [155]. Although the properties of this new fusion protein were not further studied, the authors suggest that the TRIP12-NPM protein may inhibit the function of ARF. *TRIP12* somatic mutations at splice sites were identified in pancreatic cancer recurrence of two patients but not in matched originally resected primary tumor and were qualified as subclonal mutations [156].

Altogether, these studies suggest that although the *TRIP12* gene is found mutated in various type of tumors at variable frequencies, the functional consequences of these mutations in cancer development remain largely unidentified. Possible applications to detect these mutations as diagnostic, prognostic, or predictive still remain to be demonstrated.

#### 6.2.2. Alteration of TRIP12 mRNA Expression in Cancers

If we refer to the Gepia2 database (see Section Software and databases), the *TRIP12* mRNA level compared to normal corresponding tissues varies depending on the type of cancer. Indeed, among 27 different types of cancer, the down-regulation of *TRIP12* mRNA is found in 12 cancer types compared to normal corresponding tissues but reached a statistical importance only in testicular germ cell tumors. In contrast, an overexpression of *TRIP12* mRNA is found in 15 types of cancer with a statistical significance for lymphoid neoplasm diffuse large B-cell lymphoma, thymoma and pancreatic adenocarcinoma (Figure 11). Of note, the *TRIP12* mRNA level is decreased in tumor types for which *TRIP12* mRNA is elevated in normal corresponding tissue, and inversely, it is up-regulated in tumors for which the *TRIP12* mRNA level is low in corresponding normal tissues (i.e., pancreatic adenocarcinoma). However, the mechanisms responsible for the alteration of *TRIP12* mRNA in tumor samples remain largely unknown. In certain types of cancer, a high level of *TRIP12* mRNA is associated with a shorter disease-free survival (i.e., bladder urothelial carcinoma, pancreatic adenocarcinoma), which suggests that the *TRIP12* mRNA level could be a useful prognostic marker. Additionally, RNA-seq analysis of PBMC (Peripheral Blood Mononuclear Cells) obtained from a 47-year-old female patient with Acute Myeloid Leukemia (AML) with complete remission before and after induction therapy (fludarabine, cytarabine, and G-CSF (FLAG) protocol) revealed, among others, an exon 3 skipping in the *TRIP12* gene [157]. The authors claim that the alternative splicing of *TRIP12* mRNA could inhibit the TRIP12-mediated degradation of ARF, leading to an activation of the ARF/P53 signaling pathway and an induction of apoptosis of cell with aberrant growth. This process could contribute to the remission.

#### 6.2.3. Alterations of TRIP12 Protein Expression in Cancers and Consequences

Available data on TRIP12 protein expression in cancer are much less abundant than those describing TRIP12 mRNA levels. The Human Protein Atlas indicates a heterogeneous expression of TRIP12 depending on cancer type with a high TRIP12 frequency in head-and-neck cancer, testis cancer, and glioma patients. Only a few molecular events leading to alterations of TRIP12 protein expression are known. For instance, Cai et al. demonstrated an overexpression of the deubiquitinase USP7 in hepatocellular carcinoma (HCC) tissues that is correlated with poor prognosis [47]. They showed that USP7 is required for the proliferation of HCC-derived cell lines. Mechanistically, the study reveals that USP7 overexpression stabilizes the TRIP12 protein, which in turn ubiquitinates and degrades ARF to promote HCC-derived cell proliferation. Interestingly, they established that the expressions of TRIP12 and/or USP7 are independent prognosis markers of HCC. *Helicobacter pylori* (*H. pylori*) is one of the most common human pathogens. This bacterium is recognized as the strongest identified risk factor for gastric malignancies. Horvat et al. demonstrated that infection by *H. pylori* of gastric cancer derived-cell lines induces TRIP12 protein expression through the implication of the virulence factor CagA (Cytotoxin-Associated Gene A) by a still unknown mechanism [158]. TRIP12 overexpression was observed in human gastric mucosa infected with *H. pylori*. The consequence of TRIP12-induced expression is a polyubiquitination-mediated degradation of ARF that leads to an inhibition of autophagy in gastric cells. Patients with human papillomavirus (HPV)-positive head and neck squamous cell carcinoma (HNSCC) have a better response to radiotherapy and higher overall survival rate than those with HPV-negative HNSCC. Indeed, HPV infection leads to P16 overexpression, which in turn provokes a destabilization of the TRIP12 protein by an unknown mechanism [159]. As TRIP12 is a DDR regulator (detailed in Section 5.3.), the down-regulation of TRIP12 sensitizes HPV-positive HNSCC to radiotherapy. Likewise, in 18 HPV-positive HNSCC patients, the overexpression of TRIP12 is associated with a poor survival rate. These results were confirmed recently by the same group [160]. Recently, Gatti et al. identified that the TRIP12 protein level could provide a useful marker for PARP inhibitors (PARPi) sensitivity [27]. Indeed, responses to this therapy are promising, but predictions based on PARP1 gene expression abundance and/or activity are not accurate enough. TRIP12 limits PARP1 presence at the chromatin by polyubiquitinating it and, as a consequence, it decreases the efficiency of PARP1 trapping by PARPi. Of importance, the TRIP12 protein level is negatively correlated with PARP1 protein abundance in two cohorts of breast and ovarian cancer patients.

We described in Section 5.2. the essential role of ARF in the activation of the P53 pathway and its degradation mediated by TRIP12. We also described how the nucleophosmin (NPM) protects ARF from TRIP12-mediated degradation by sequestrating it into the nucleoli [15]. The NPM gene is mutated in one-third of Acute myeloid leukemia (AML), which provokes the translocation of NPM protein into the cytoplasm and the loss of its capacity to maintain ARF localization in the nucleolus [96]. Combined with an overexpression of TRIP12, a frequent loss of NPM nucleolar localization strongly impedes the activation of the ARF/P53 pathway in AML-derived cells and favors their proliferation.

## 7. Conclusions and Perspectives

In this review, we provide a synthesis of the different knowledge on the E3 ubiquitin ligase TRIP12. First described as a thyroid hormone receptor interacting protein, it appears that the TRIP12 protein displays several important functions in essential cellular processes such as cell cycle progression, DNA damage repair, chromatin remodeling, and cell differentiation. Although the number of studies related to this E3 ubiquitin ligase is constantly growing, numerous aspects of its regulation and functions in physiology and pathologies are not entirely clear.

High-throughput sequencing analyses have determined the level of expression of *TRIP12* mRNA in a myriad of healthy and pathological human tissues such as cancers. However, the molecular mechanisms that regulate the production and the stability of *TRIP12* mRNA remain largely unknown. The same observation can be made for the post-translational regulation of TRIP12. Indeed, mass spectrometry approaches identified numerous post-translationally modified residues in TRIP12. Unfortunately, none of them have been linked to biological functions, even if three phosphorylation sites are attributed to specific kinases involved in cell cycle (CDK1) and DNA damage response (ATR/ATM, CHK1). We can speculate that these phosphorylated residues modify the protein-interacting properties or the ubiquitin ligase capacity of TRIP12, which in turn participates in cell cycle control and DDR. We could also envisage the possibility that TRIP12 phosphorylated residues result from the activity of kinases implicated in other signaling pathways (i.e., MAP kinases, PI3 kinases). The development of antibodies specific to TRIP12-phosphorylated residues is crucial to decipher the biological consequences of these post-translational modifications. Likewise, the identification of the ubiquitin ligase(s) responsible for the ubiquitinated residues of TRIP12 would certainly help to better perceive the regulation of the TRIP12 protein during the cell cycle. The anaphase-promoting complex/cyclosome (APC/C) system ubiquitinates several cell cycle-regulated proteins early after mitosis (i.e., CYCLIN A and B), thereby mediating degradation by the proteasome. TRIP12 possesses putative KEN box (aa 1496–1570/UniProt source) and Destruction boxes (aa 859–867 and aa 1546–1554/ELM resource) specifically recognized by the APC/C, system making this degradation system a legitimate candidate for the control of TRIP12 stability. To confirm these multiple hypotheses, a complete and exhaustive interactome/substratome of TRIP12 is required, keeping in mind that due to its ubiquitous expression, the interactome of TRIP12 will possibly vary depending on the cell type. We can logically expect other important cellular functions for this E3 ubiquitin ligase depending on the cell type or the cellular context.

This review raises questions, one of which is whether TRIP12 is an appropriate therapeutic target. In the case of cancers in which TRIP12 is overexpressed (i.e., pancreatic cancer) and given its role in the maintenance of a low level of P53 mediated by ARF degradation, it seems logical that the inhibition of TRIP12 could be a promising therapeutic strategy. The inhibition of TRIP12 ubiquitin ligase activity or its interaction with protein partners might counteract the proliferation rate of cancer cells or sensitize them to chemotherapies (by affecting DNA damages repair induced by chemotherapies). To this purpose, the crystallization of the entire TRIP12 protein is a prerequisite to obtain the 3D structure of the protein to facilitate the screening and/or the design of specific inhibitors. PROTACs that are bifunctional molecules made up of an E3 ubiquitin ligase and a target protein connected via a linker could also be used to target TRIP12.

In pathological contexts where the *TRIP12* gene is mutated or truncated (numerous types of cancer and intellectual disorders), we do not yet understand whether the TRIP12 protein is indeed expressed or if the mRNA is degraded. If expressed, the functional consequences of mutated and truncated forms of TRIP12 in these pathologies are still unknown. We can only speculate that mutant proteins would act as dominant negative proteins, leading to an alteration of genome expression. The development of murine transgenic models expressing *Trip12* mutated forms could address these issues. Moreover, several truncating mutations identified in cancers and IDs theoretically lead to TRIP12 proteins lacking the HECT catalytic domain or protein–protein interacting domains (WWE/ARM). Therefore, therapeutic strategies inhibiting its catalytic activity or protein interaction might be useless or, even worse, deleterious for the patients.

To conclude, the TRIP12 protein is emerging as an important protein in several essential cellular processes as well as in pathological alterations. Uncovering the function of TRIP12 offers a glimpse for potential exciting therapeutic strategies. However, numerous aspects of its regulations and functions need to be better described prior to its application in therapeutic objectives.

## 8. Software and Data Bases

**BioGrid:** The Biological General Repository for Interaction Datasets (BioGrid) is an open access database that houses genetic and protein interactions curated from the primary biomedical literature for all major model organism species and humans (https://thebiogrid.org/).

**cBioPortal:** The cBio Cancer Genomics Portal is an open-access resource for the interactive exploration of multidimensional cancer genomics datasets, currently providing access to data from more than 5000 tumor samples from 20 cancer studies (www.cbioportal.org).

**ClinVar:** ClinVar is a public database of variant interpretations that has steadily grown to become the largest publicly available genetic variant database and provides an ever-growing resource to study genotype–phenotype correlations (https://www.ncbi.nlm.nih.gov/clinvar/).

**COBALT:** The constraint-based Multiple Alignment Tool is a multiple sequence alignment tool that finds a collection of pairwise constraints derived from conserved domain database, protein motif database, and sequence similarity, using RPS-BLAST, BLASTP, and PHI-BLAST. Then, pairwise constraints are incorporated into a progressive multiple alignment (www.ncbi.nlm.nih.gov/tools/cobalt/cobalt.cgi).

**EMBOSS matcher:** EMBOSS Matcher identifies local similarities in two input sequences using a rigorous algorithm based on Bill Pearson’s lalign application (www.ebi.ac.uk).

**ELM:** The Eukaryotic Linear Motif resource is a computational biology resource for investigating short linear motifs (SLiMs) in eukaryotic proteins. It is currently the largest collection of linear motif classes with annotated and experimentally validated linear motif instances (http://elm.eu.org/).

**Genome browser:** A genome browser is a graphical interface for the display of information from a biological database for genomic data (www.genome.ucsc.edu).

**Gepia2:** The Gene Expression Profiling Interactive Analysis 2 web server is a valuable and highly cited resource for gene expression analysis based on tumor and normal samples from the TCGA and the GTEx databases (http://gepia2.cancer-pku.cn).

**Human Protein Atlas:** The Human Protein Atlas is a program with the aim to map all the human proteins in cells, tissues, and organs using an integration of various omics technologies, including antibody-based imaging, mass spectrometry-based proteomics, transcriptomics, and systems biology (www.proteinatlas.org).

**IDEAL:** Intrinsically Disordered proteins with Extensive Annotations and Literature is a collection of knowledge on experimentally verified intrinsically disordered proteins. IDEAL contains manual annotations by curators on intrinsically disordered regions, interaction regions to other molecules, post-translational modification sites, references, and structural domain assignments (www.ideal-db.org).

**InterProScan:** InterProScan is a tool that scans given protein sequences against the protein signatures of the InterPro member databases, currently—PROSITE, PRINTS, Pfam, ProDom, and SMART (https://www.ebi.ac.uk/interpro/).

**IUPred:** Intrinsically unstructured/disordered proteins have no single well-defined tertiary structure in their native, functional state. This server recognizes such regions from the amino acid sequence based on the estimated pairwise energy content (https://iupred.elte.hu/).

**LOVD:** The Leiden Variation Database is a free, flexible web-based open source database developed in the Leiden University Medical Center in the Netherlands, which is designed to collect and display variants in the DNA sequence. The focus of an LOVD is usually the combination between a gene and a genetic (heritable) disease (https://www.lovd.nl/).

**OMIM: Online Mendelian Inheritance in Man** is a comprehensive, authoritative compendium of human genes and genetic phenotypes that is freely available and updated daily (https://www.omim.org/).

**PhosphoSitePlus^®^:** PhosphoSitePlus^®^ provides comprehensive information and tools for the study of protein post-translational modifications (PTMs) including phosphorylation, acetylation, and more (https://phosphosite.org/).

**SFARI GENE:** SFARI GENE is an evolving database for the autism research community that is centered on genes implicated in autism susceptibility (https://gene.sfari.org/).

**STRING:** Search Tool for the Retrieval of Interacting Genes/Proteins is a biological database and web resource of known and predicted protein–protein interactions. The STRING database contains information from numerous sources, including experimental data, computational prediction methods, and public text collections. It is freely accessible and it is regularly updated (https://string-db.org/).

**UniProt:** UniProt is the Universal Protein resource, a central repository of protein data created by combining the Swiss-Prot, TrEMBL and PIR-PSD databases. (www.uniprot.org).

## Figures and Tables

**Figure 1 ijms-21-08515-f001:**
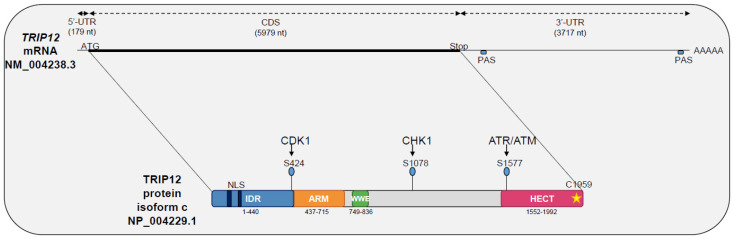
Schematic representation of Thyroid hormone Receptor Interacting Protein 12 (*TRIP12*) mRNA (NM_004238.3) and TRIP12 protein (isoform c, NP_004229.1) with functional domains, kinase associated-phosphorylated residues and cysteine catalytic site (yellow star). UTR: Untranslated Region, CDS: Coding Sequence, PAS: Poly-Adenylation Signal, AAA: Poly-adenylated tail, ATG: translation Start codon, Stop: translation Stop codon, NLS: Nuclear Localization Signal, S: serine residue, aa: amino acid, IDR: Intrinsically Disordered Regions, ARM: Armadillo repeats, HECT: Homologous to the E6-AP C-Terminus.

**Figure 2 ijms-21-08515-f002:**
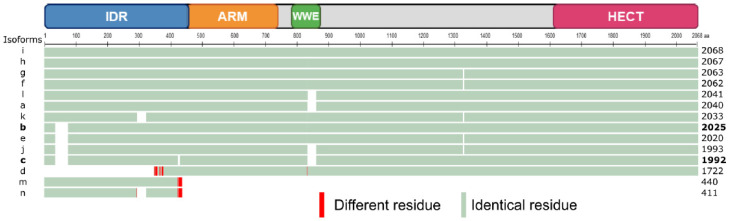
Schematic representation of a multiple alignment of human TRIP12 isoforms using the COBALT multiple alignment tool (see Section Software and databases). TRIP12 isoform i (2068 aa) was defined as the sequence of reference (master sequence). The two most studied isoforms b and c (and size) are written in bold. Residues in green are identical and in red are different. The different domains of TRIP12 are indicated above the alignment.

**Figure 3 ijms-21-08515-f003:**
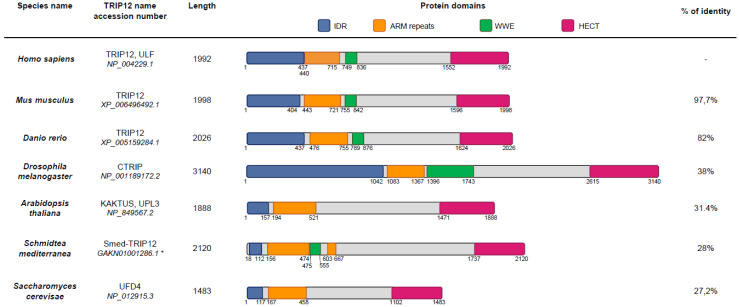
Schematic representation of protein domains in TRIP12 orthologues. Domain boundaries were defined using InterProScan analysis. Percentage of identity was determined by Emboss matcher (see Section Software and databases) pairwise sequence alignment (using *Homo sapiens* TRIP12 protein sequence as reference [43]). *: *Schmidtea mediterranea* cDNA sequence was manually corrected to perform the analysis.

**Figure 4 ijms-21-08515-f004:**
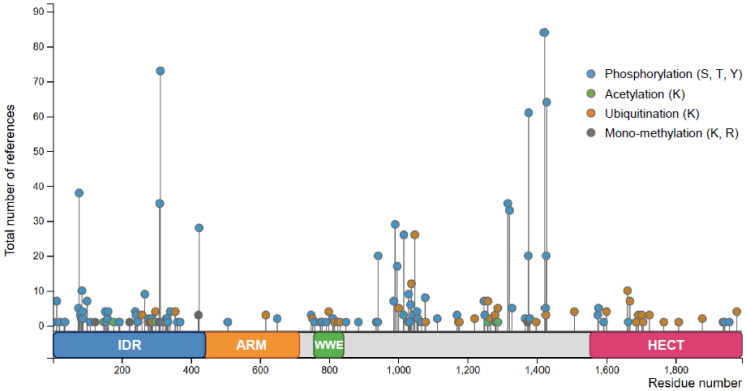
Post-translational modifications of TRIP12 (isoform c) adapted from PhosphoSitePlus^®^ (see Section Software and databases ). Total number of references represents the number of records in which this modification site was assigned using proteomic discovery mass spectrometry and using other methods. S stands for serine, T stands for threonine, Y stands for tyrosine, and K stands for lysine.

**Figure 5 ijms-21-08515-f005:**
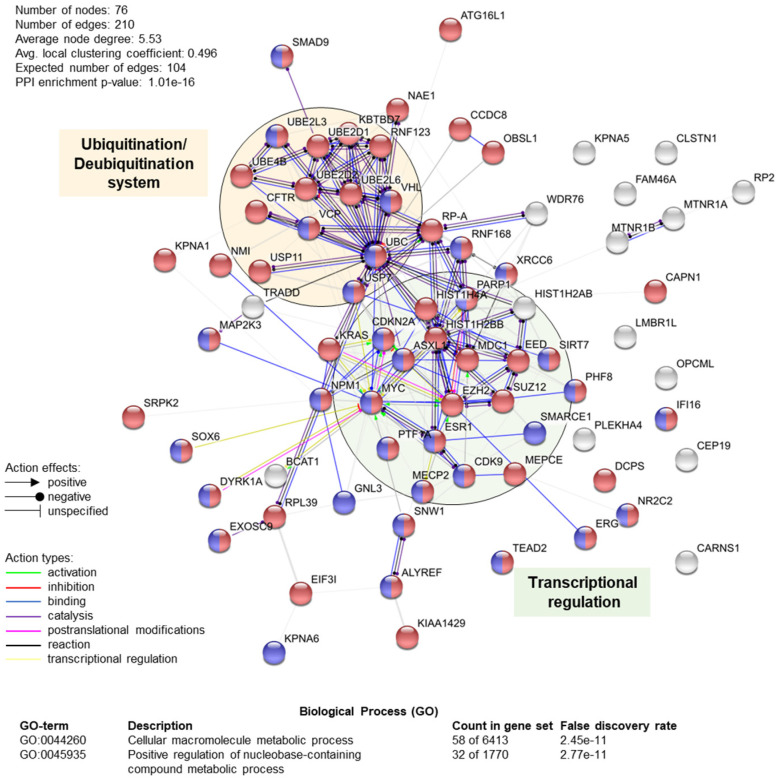
Schematic representation of human TRIP12-interacting protein–protein network (*n* = 76) and functional enrichment analysis using STRING software (see Section Software and databases). Network nodes represent proteins. Red nodes represent proteins belonging to cellular macromolecule metabolic process (GO term: 0044260), and purple nodes represent proteins belonging to the positive regulation of the nucleobase-containing compound metabolic process (GO-term: 0045935). Edges represent protein–protein molecular actions (indicated in the figure). Line shape indicates the predicted mode of action (indicated in the figure). Active interaction sources: text mining, experiments, databases, co-expression, neighborhood and co-occurrence. Minimum required interaction score: medium confidence (0.400). Green and orange filled circles indicate proteins implicated in “transcriptional regulation” and “ubiquitination/deubiquitination system”, respectively.

**Figure 6 ijms-21-08515-f006:**
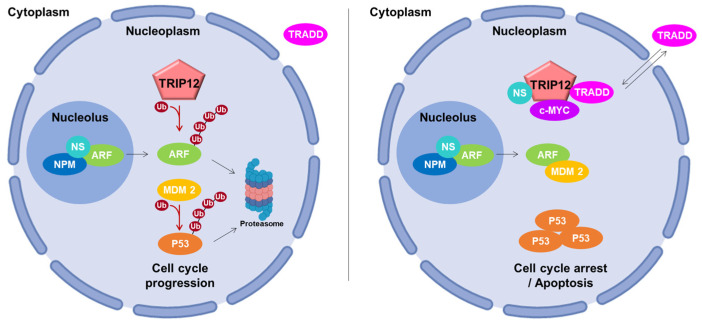
Schematic representation of TRIP12, ARF, and P53 pathway. On the left, TRIP12 controls ARF and indirectly P53 ubiquitination degradation by proteasome, while TRIP12 participates in cell cycle progression [58,75,96,115]. On the right, ARF is protected from TRIP12 when bound to nucleophosmin (NPM)/nucleostemin (NS) and localized in the nucleolus. Moreover, in the nucleoplasm, TRIP12 is sequestered by different proteins (in the presence of NS or TRADD and in high level c-MYC condition), blocking its interaction with ARF. ARF binds and inhibits MDM2, which leads to P53 accumulation and cell cycle arrest.

**Figure 7 ijms-21-08515-f007:**
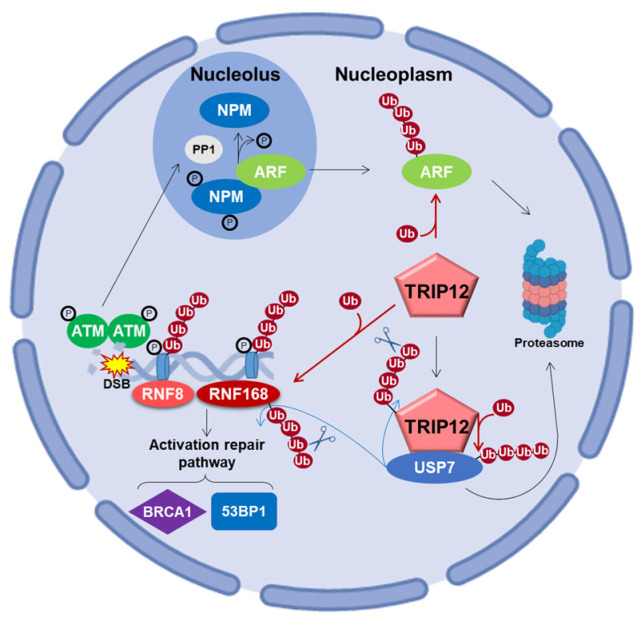
Schematic representation of TRIP12 in the DNA damage repair (DDR) pathway in response to DNA double-strand break (DSB) [14,39,115,118,119,120,121]. In response to double-strand breaks (DSB), ATM, RNF8, and RNF168 are recruited on the damage and trigger the propagation of the ubiquitination of H2A and H2AX far from the initial DNA lesions. TRIP12 controls the nuclear pool of RNF168 for the ubiquitination of chromatin after DNA disruption. Indeed, TRIP12 ubiquitinates and induces the degradation of RNF168 by the proteasome. TRIP12 controls Ubiquitin-Specific Peptidase 7 (USP7) ubiquitination, which in turn stabilizes it.

**Figure 8 ijms-21-08515-f008:**
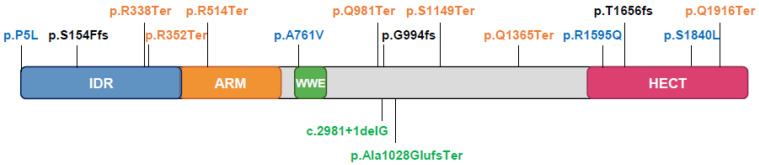
Schematic illustration of published *TRIP12* gene mutations in intellectual disorders. Mutations in blue are missense, mutations in orange are nonsense, mutations in black are frameshift. Mutations in green are located in splice donor sites. From [10,142,143,144,145].

**Figure 9 ijms-21-08515-f009:**
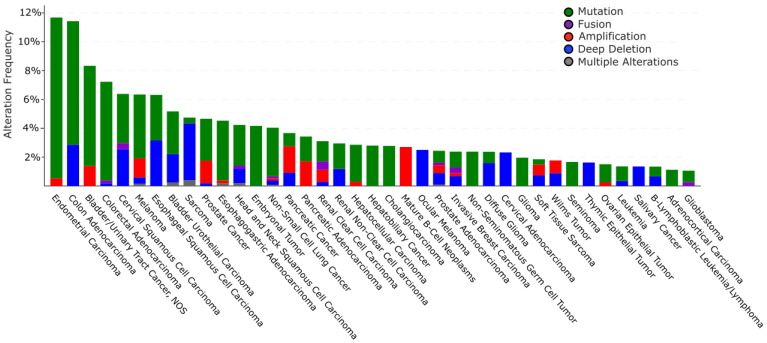
Copy number alterations (CNA) and mutations of the *TRIP12* gene in human cancers in cBioPortal (see Section Software and databases) obtained from 176 non-redundant studies containing more than 100 cases with both mutations and CNA data (27,235 patients and 28,253 samples). Only cancer types with more than 1% of altered cases are represented.

**Figure 10 ijms-21-08515-f010:**
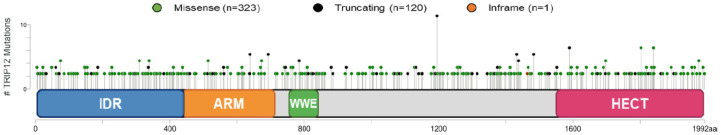
Schematic representation of mutations on the *TRIP12* gene in human cancers in cBioPortal (see Section Software and databases) obtained from 176 non-redundant studies containing more than 100 cases with both mutations and CNA data (27,235 patients and 28,253 samples). Mutation diagram circles are colored with respect to the corresponding mutation types indicated in the legend. In case of different mutation types at a single position, the color of the circle is determined with respect to the most frequent mutation type. Truncating mutations include nonsense, nonstop, frameshift deletion, frameshift insertion, and splice site. Inframe mutations include inframe deletion and inframe insertion.

**Figure 11 ijms-21-08515-f011:**
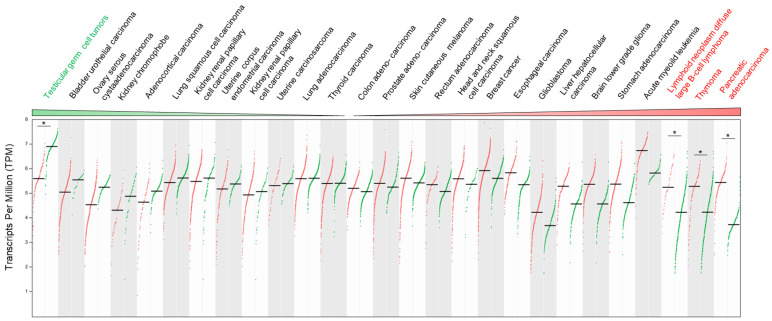
*TRIP12* mRNA expression level in 27 different types of cancer adapted from Gepia2 website (see Section Software and databases). Only tumor/normal tissue matched pairs with more than 25 cases were considered. Each dot represents a normal (green) or a tumor (red) sample. The horizontal black bar represents the mean. The *TRIP12* mRNA level was obtained by RNA-seq and expressed as transcripts per million (TPM). Cancer types in which *TRIP12* mRNA is statistically differentially expressed (ANOVA test with a *q* value < 0.01) are marked by an asterisk.

**Table 1 ijms-21-08515-t001:** List of the human mRNA variants and protein isoforms of TRIP12.

mRNA Variant	Accession Number	mRNA Length	Exon Number	Protein Isoform	Protein ID	aa Number	Theoretical kDa
**1**	NM_001284214.2	10019	42	a	NP_001271143.1	2040	224
**2**	NM_001284215.2	9974	41	b	NP_001271144.1	2025	223
**3**	NM_004238.3	9875	41	c	NP_004229.1	1992	219
**4**	NM_001284216.2	9065	39	d	NP_001271145.1	1722	189
**5**	NM_001348315.2	10408	42	a	NP_001335244.1	2040	224
**6**	NM_001348316.2	10070	41	b	NP_001335245.1	2025	223
**7**	NM_001348317.1	9959	41	e	NP_001335246.1	2020	222
**8**	NM_001348318.2	10348	41	e	NP_001335247.1	2020	222
**9**	NM_001348319.1	10085	42	f	NP_001335248.1	2062	227
**10**	NM_001348320.2	10474	42	f	NP_001335249.1	2062	227
**11**	NM_001348321.1	10088	42	g	NP_001335250.1	2063	227
**12**	NM_001348322.1	10218	43	h	NP_001335251.1	2067	227
**13**	NM_001348323.3	10100	42	h	NP_001335252.1	2067	227
**14**	NM_001348324.2	10196	42	h	NP_001335253.1	2067	227
**15**	NM_001348325.2	10611	43	h	NP_001335254.1	2067	227
**16**	NM_001348326.2	10486	42	h	NP_001335255.1	2067	227
**17**	NM_001348327.2	10489	42	h	NP_001335256.1	2067	227
**18**	NM_001348328.1	10103	42	i	NP_001335257.1	2068	227
**19**	NM_001348329.2	10454	42	i	NP_001335258.1	2068	227
**20**	NM_001348330.2	10492	42	i	NP_001335259.1	2068	227
**21**	NM_001348331.1	9878	41	j	NP_001335260.1	1993	219
**22**	NM_001348332.1	9998	43	k	NP_001335261.1	2033	224
**23**	NM_001348333.1	10022	42	l	NP_001335262.1	2041	225
**24**	NM_001348335.1	5955	6	m	NP_001335264.1	440	48
**25**	NM_001348336.1	6052	6	m	NP_001335265.1	440	48
**26**	NM_001348334.1	5951	7	n	NP_001335263.1	411	45

**Table 2 ijms-21-08515-t002:** List of the different protein interactors of human TRIP12.

#	Interactor Gene	Interactor Name	Bait and/or	Interaction Detection Method	Interaction Type	References
			Hit			
**1**	ALYREF	Aly/REF export factor	bait	aff tech	co-localization	[66]
**2**	**ASXL1**	**Additional Sex Combs-Like Protein 1**	bait	biochemical	ph ass	[53]
**3**	ATG16L1	Autophagy-related protein 16 like1	bait	aff tech	ph ass	[67]
**4**	BCAT1	Branched-chain-amino-acid aminotransferase	bait	ph ass	ph ass	[68,69]
**5**	CAPN1	Calpain-1 catalytic subunit	bait	aff tech	ph ass	[69]
**6**	CARNS1	Carnosine synthase 1	bait	two hybrid	dir int	[70]
**7**	CCDC8	Coiled-coil domain-containing protein 8	bait	aff tech	ph ass	[71]
**8**	CDK9	Cyclin-dependent kinase 9	bait, hit	aff tech	ph ass	[72]
**9**	**CDKN2A**	**Cyclin-Dependent Kinase Inhibitor 2A/p14ARF**	bait	aff tech, pd, ph ass	ph ass, dir int, ph int	[15,58,73,74,75]
**10**	CEP19	Centrosomal protein of 19 kDa	bait	bioid	ph ass	[76]
**11**	CFTR	Cystic fibrosis transmembrane conductance regulator	bait	aff tech	ph ass	[77]
**12**	CLSTN1	Calsyntenin-1	bait	two hybrid	dir int	[70]
**13**	DCPS	Decapping scavenger enzyme	bait	aff tech	ph ass	[78]
**14**	DYRK1A	Dual specificity Tyrosine-phosphorylation-regulated kinase 1A	bait	aff tech	ph ass	[79]
**15**	EED	Embryonic ectoderm development protein	bait	aff tech	ph ass	[80]
**16**	EIF3I	Eukaryotic translation initiation factor 3 subunit I	bait	aff tech	ph ass	[68,69]
**17**	ERG	Transcriptional regulator ERG	bait	pd	dir int	[81]
**18**	ESR1	Estrogen receptor alpha	bait	aff tech	ph ass	[82]
**19**	EXOSC9	Exosome complex component RRP45	bait	aff tech	ph ass	[83]
**20**	EZH2	Histone-lysine N-methyltransferase	bait	aff tech	ph ass	[84]
**21**	FAM46A	HBV X-transactivated gene 11 protein)	bait	aff tech	ph ass	[83]
**22**	GNL3	Guanine nucleotide-binding protein-like 3/Nucleostemin	hit	aff tech	ph ass, dir int, ph int	[75]
**23**	HIST1H2AB	Histone H2A	bait	aff tech	ph ass	[85]
**24**	HIST1H2BB	Histone H2B	bait	aff tech, bioid	ph ass	[85,86]
**25**	HIST1H4A	Histone H4	bait	aff tech	ph ass	[85,87]
**26**	IFI16	Gamma-interferon-inducible protein 16	bait	aff tech	ph ass	[88]
**27**	KBTBD7	Kelch repeat and BTB domain-containing protein 7	bait	aff tech	ph ass	[69]
**28**	KIAA1429	Protein virilizer homolog	bait	aff tech	ph ass	[89]
**29**	KPNA1	Importin subunit alpha-5 (Karyopherin subunit alpha-1)	hit	aff tech	ph ass	[90]
**30**	KPNA5	Importin subunit alpha-6 (Karyopherin subunit alpha-5)	hit	aff tech	ph ass	[90]
**31**	KPNA6	Importin subunit alpha-7 (Karyopherin subunit alpha-6)	hit	aff tech	ph ass	[90]
**32**	KRAS	GTPase KRas	bait	aff tech	ph ass	[91]
**33**	LMBR1L	Limb development membrane protein 1-like	bait	aff tech	ph ass	[92]
**34**	MAP2K3	Dual specificity mitogen-activated protein kinase kinase 3	bait	aff tech	ph ass	[91]
**35**	MDC1	Mediator of DNA damage checkpoint protein 1	bait	aff tech	ph ass	[93]
**36**	MECP2	Methyl-CpG-binding protein 2	bait	aff tech	ph ass	[69]
**37**	MEPCE	7SK snRNA methylphosphate capping enzyme	bait	aff tech	ph ass	[72]
**38**	MTNR1A	Melatonin receptor type 1A	bait	two hybrid	dir int	[94]
**39**	MTNR1B	Melatonin receptor type 1B	bait	two hybrid	dir int	[94]
**40**	MYC	Myc proto-oncogene protein	bait, hit	aff tech, pd	ph ass, dir int	[15,95]
**41**	**NAE1**	**NEDD8-activating enzyme E1 regulatory subunit/APP-BP1**	hit	aff tech, enz study	ph ass, dir int	[4]
**42**	NMI	N-myc-interactor	bait, hit	aff tech, pd	ph ass, dir int	[74]
**43**	NPM	Nucleophosmin	bait, hit	aff tech, pd	ph ass, dir int	[96]
**44**	NR2C2	Nuclear receptor subfamily 2 group C member 2	bait	aff tech	ph ass	[97]
**45**	OBSL1	Obscurin-like protein 1	bait	aff tech	ph ass	[71]
**46**	OPCML	Opioid-binding protein/cell adhesion molecule	bait	aff tech	ph ass	[69]
**47**	**PARP1**	**Poly-(ADP Ribose) polymerase 1**	bait, hit	aff tech, pd, enz study	ph ass, dir int	[27]
**48**	PHF8	Histone lysine demethylase	bait	aff tech	ph ass	[83]
**49**	PLEKHA4	Peckstrin homology domain containing, family member 4	bait	aff tech	ph ass	[98]
**50**	**PTF1A**	**Pancreas transcription factor 1 subunit alpha**	bait, hit	aff tech, FRET, two hybrid	ph ass, dir int, dir int	[6]
**51**	RNF123	E3 ring finger ubiquitin ligase 123	bait	aff tech	ph ass	[99]
**52**	**RNF168**	**E3 ring finger ubiquitin ligase168**	bait	aff tech	ph ass	[14]
**53**	RP2	Protein XRP2	bait	aff tech	ph ass	[68,69]
**54**	RPA	Replication protein A	bait	aff tech	ph ass	[100]
**55**	RPL39	60S ribosomal protein L39	hit	aff tech	ph ass	[90]
**56**	SIRT7	Sirtuin 7	bait	aff tech	ph ass	[101]
**57**	SMAD9	SMAD family member 9	bait	two hybrid	dir int	[102]
**58**	**SMARCE1**	**SWI/SNF-related matrix-associated actin-dependent regulator of chromatin subfamily E member 1/BAF57**	bait	aff tech	ph ass	[54]
**59**	SNW1	SNW domain-containing protein 1	bait	aff tech	ph ass	[103]
**60**	**SOX6**	**Transcription factor SOX-6**	bait, hit	enz study, aff tech, two hybrid	dir int, ph ass	[12]
**61**	SRPK2	SRSF protein kinase 2	bait	enz study	dir int	[104]
**62**	SUZ12	Polycomb protein SUZ12	bait	aff tech	ph ass	[84]
**63**	TEAD2	Transcriptional enhancer factor TEF-4	bait	aff tech	ph ass	[105]
**64**	TRADD	Tumor necrosis factor receptor type 1-associated DEATH domain	bait, hit	aff tech	ph ass	[58]
**65**	UBC	Ubiquitin-C	bait, hit	enz study, aff tech	dir int, ph ass	[60,106,107]
**66**	UBE2D1	Ubiquitin-conjugating enzyme E2 D1	hit	pd	dir int	[4,12,60,108]
**67**	UBE2D2	Ubiquitin-conjugating enzyme E2 D2	bait	pd	dir int	[109]
**68**	UBE2L3	Ubiquitin-conjugating enzyme E2 L3	bait, hit	pd	dir int	[108,109]
**69**	UBE2L6	Ubiquitin-conjugating enzyme E2 L6	bait	aff tech	ph ass	[91]
**70**	UBE4B	Ubiquitin conjugation factor E4 B	bait	pd	dir int	[4]
**71**	**USP7**	**Ubiquitin-specific-peptidase 7**	bait, hit	aff tech, ph ass	ph ass	[39,48,110,111]
**72**	USP11	Ubiquitin-specific peptidase 11	bait	aff tech	ph ass	[110]
**73**	VCP	Valosin-containing protein	bait	aff tech	ph ass	[112]
**74**	VHL	von Hippel-Lindau E3 ubiquitin ligase	bait	aff tech	ph ass	[83]
**75**	WDR76	WD repeat-containing protein 76	bait	aff tech	ph ass	[85]
**76**	XRCC6	X-ray repair cross-complementing protein	bait	bioid	ph ass	[113]

Human TRIP12 protein interactors based on Biogrid database (see Section Software and databases) and the literature. Interactors in bold (*n* = 9) were further characterized as TRIP12 ubiquitinated substrates. Proteins used to capture protein complexes in which TRIP12 was identified were designated as “bait”. Hits are proteins detected only in the immunoprecipitation (IP) of the TRIP12 protein but not in a vector control IP. Aff tech: affinity techniques, bioid: biotin identification, dir int: direct interaction, enz study: enzymatic study, FRET: fluorescence resonance energy transfer, pd: pull down, ph ass: physical association and ph int: physical interact.

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
