# Peer review of "E3 Ubiquitin Ligase TRIP12: Regulation, Structure, and Physiopathological Functions"

_ijms, 2020, doi:10.3390/ijms21228515_

Round 1

Reviewer 1 Report

Some typographical errors that change the meaning within the text should be corrected. For e.g.

Page 12: "wild opens" should be replaced by "wide open"

Page 12: "binding partners several"; delete the word "several"

Page 19: "ASXL1 et BAP1" should be replaced by "ASXL1 and BAP1"

Page 20: "acinar specific genes for such as the amylase"; delete the words "for" and "the" from this sentence

Page 22: "Punctual mutations" same as "point mutations?" Then replace with latter (more common).

Reword the awkward concluding sentence "a synthesis of the different knowledge-base surrounding the E3s" 

Author Response

Some typographical errors that change the meaning within the text should be corrected.

Response: A careful reading of the manuscript was performed.

For e.g.

Page 12: "wild opens" should be replaced by "wide open"

Response: The error was corrected.

Page 12: "binding partners several"; delete the word "several"

Response: the word “several” was deleted.

Page 12: "ASXL1 et BAP1" should be replaced by "ASXL1 and BAP1"

Response: The error was corrected.

Page 12: "acinar specific genes for such as the amylase"; delete the words "for" and "the" from this sentence

Response: the words “the” and “for” were deleted.

Page 14: "Punctual mutations" same as "point mutations?" Then replace with latter (more common).

Response: the word “punctual” was replaced by “point”.

Reword the awkward concluding sentence "a synthesis of the different knowledge-base surrounding the E3s" 

Response: the sentence was replaced by “ a synthesis of the different knowledge on the E3 ubiquitin ligase TRIP12”.

Reviewer 2 Report

Brunet et al. summarize literature on TRIP12, a HECT E3 ubiquitin ligase. The contents are extensive to cover from genetic structure to potential involvement in diseases. Despite the extensive contents on TRIP12, the context of the current manuscript does not seem to be classified as a "review", which raises a serious concern whether the current manuscript is qualified to be published in this journal.

Major concerns:

The reviewer is NOT convinced of the biological significance of TRIP12 as a topic for a separate review at this time. It seems that the functions and mechanisms of TRIP12 in various biological processes have NOT been firmly established, as the authors acknowledged at multiple instances through the manuscript. The manuscript contains too much speculations, not summary of mechanistic and functional research activities reflected in the literature. Also not convincing is why TRIP12, not any other member of "TRIP" family proteins, is chosen for the topic of this manuscript. The reviewer is concerned that the audience would be benefitted very little from the current manuscript.

Specific concerns:

  1. The manuscript is too descriptive: the authors rely too much on outputs of various databases, many of the items related to TRIP12 not being confirmed mechanistically nor functionally. Table 2, Figure 2, Figure 4, Table 3, Figure 8, Figure 9, Figure 10 and Figure 11simly list up the outputs from various databases without any biological insight. 
  2. The authors are encouraged to check the usage of capital letters in words appearing in the middle of sentences: e.g., Vertebrate (should've been vertebrate), Human (-> human), Mammals (-> mammals), to name a few. The reviewer also found various cases of grammatical errors such as omitting a space between words, duplication of references and so on.
  3. Table 1: no indication of two "groups" of TRIP protein family.
  4. Table 2: no "NM_001284215.1" in the table.
  5. Figure 2: alternatively spliced forms should be called as "long" and "short", NOT "longest" and "shortest".
  6. Some sentences are bold; the reviewer does not understand why some sentences are shown in bold.

Author Response

Brunet et al. summarize literature on TRIP12, a HECT E3 ubiquitin ligase. The contents are extensive to cover from genetic structure to potential involvement in diseases. Despite the extensive contents on TRIP12, the context of the current manuscript does not seem to be classified as a "review", which raises a serious concern whether the current manuscript is qualified to be published in this journal.

Response: The aim of this review is to compile the different knowledge regarding the E3 ubiquitin ligase TRIP12. This knowledge was found in the literature as well as in the public resources (BioGrid, Clustal Omega, NCBI, etc…). None of the Figures and Tables in this review corresponds to original results. To our opinion, this manuscript cannot be classified as an original Article.

Major concerns:

The reviewer is NOT convinced of the biological significance of TRIP12 as a topic for a separate review at this time. It seems that the functions and mechanisms of TRIP12 in various biological processes have NOT been firmly established, as the authors acknowledged at multiple instances through the manuscript. The manuscript contains too much speculations, not summary of mechanistic and functional research activities reflected in the literature. Also not convincing is why TRIP12, not any other member of "TRIP" family proteins, is chosen for the topic of this manuscript. The reviewer is concerned that the audience would be benefitted very little from the current manuscript.

Response: We agree with Reviewer 2 that further investigations would be necessary to clarify certain aspects of physiological functions of TRIP12 depending on several identified interactors. However, we do not agree that all the functions and mechanisms of TRIP12 have not been firmly established. The role of TRIP12 in the ubiquitination of a significant number of substrates is firmly established with impacts on cell cycle progression, DNA damage response, histone ubiquitination, chromatin remodeling and cell differentiation. These biological functions are demonstrated and published in original articles cited in the paragraph 5- “The physio-pathological roles of TRIP12” published in Oncogene, Mol Cell, Nat Cell Biol, JBC, Hepatology, Nature, Cell, etc… The importance of TRIP12 in pathologies is also emerging: TRIP12 gene is now recognized as a genuine-disease causative gene for intellectual disability and autism spectrum disorder (§6.1). Consequences of alteration of TRIP12 protein expression on cancer therapies are also now demonstrated (§6.2.). Therefore, we are convinced that the objective of a review is not only to synthesize biological facts on a specific subject but also to provide different interpretations of results. Therefore, a review on TRIP12 will help to develop new ideas, research directions and perspectives that will stimulate additional functional discoveries.

There are two reasons why TRIP12 was chosen for the topic of this manuscript. Firstly, we discovered that TRIP12 is implicated in pancreatic carcinogenesis. Secondly, the number of publications and data on TRIP12 is increasing, this prompted us to synthetize them in a review, which will be the first review dedicated to this E3 ubiquitin ligase.

Specific concerns

  1. The manuscript is too descriptive: the authors rely too much on outputs of various databases, many of the items related to TRIP12 not being confirmed mechanistically nor functionally. Table 2, Figure 2, Figure 4, Table 3, Figure 8, Figure 9, Figure 10 and Figure 11 simply list up the outputs from various databases without any biological insight. 

Response: The goal of this review was to provide an exhaustive listing of the different data related to this E3 ubiquitin ligase. In other terms, we wanted to provide to IJMS readers a “what do you know about this protein”. It is obvious that all the data present in the Figures and Tables do not have YET biological significance. However, it gives to IJMS’s readers multiple directions for their research on this multifunctional protein. Please note that data presented in Fig. 8 represent published TRIP12 gene mutations in intellectual disorders, references of these publications were added to the Fig. 8 legend.

  1. The authors are encouraged to check the usage of capital letters in words appearing in the middle of sentences: e.g., Vertebrate (should've been vertebrate), Human (-> human), Mammals (-> mammals), to name a few. The reviewer also found various cases of grammatical errors such as omitting a space between words, duplication of references and so on.

Response: We carefully checked the grammatical errors and typos in the manuscript. Capital letters were removed from Vertebrates, Human, etc...

  1. Table 1: no indication of two "groups" of TRIP protein family.

Response: We agree with Reviewer 2. Table 1 does not describe two groups of TRIP proteins, the wording of the second paragraph of the introduction was confusing. We rephrased this part of the manuscript.

  1. Table 2: no "NM_001284215.1" in the table.

Response: In NCBI public data bank, the sequence NM_001284215.1 has been updated by the sequence NM_001284215.2.

  1. Figure 2: alternatively spliced forms should be called as "long" and "short", NOT "longest" and "shortest".

Response: The terms “longest” and “shortest” were replaced by “long” and “short”, respectively.

  1. Some sentences are bold; the reviewer does not understand why some sentences are shown in bold.

Response: We went through the manuscript and removed the sentences in bold.

Reviewer 3 Report

The review by Brunet and collaborators presents an overview of what is known about the E3 ligase TRIP12 (mRNA expression, regulation, location, structure and functions). The authors have done a colossal work to bring together different information about the TRIP12 protein, that were present both in the literature and in different databases. It should be noted that as a result, this manuscript is a mix between a review and a research article.

Major points

  1. Important data are missing in the introduction part. Indeed, the authors should explain, in this section, what is the interest/importance of the TRIP12 protein and therefore the interest in studying it. They should also explain the aim of their review.

  1. The authors have to cite more often the references on which their claims are based. For instance, in Table 1, references regarding the functions of the different TRIP are missing.

Other examples:

  • lane 76 “…pathway where the short form is more efficiently translated”
  • lane 188 “…the Super Elongation Complex (i.e.:AF9, AF4/FMR2)”
  • in the paragraph 4.1 there is no reference.
  • In the paragraph 4.2, between lanes 288 and 302 the authors have to add missing references in several sentences; for instance, lane 291 “…exception of APP-BP1 that is monoubiquitinated.”

… etc…

The authors should also add references in the legends.

  1. The authors should explain what software or databases they have used and not just give their names. What does the software do or what information does it contain? (IUPred, IDEAL, InterProScan, BioGrid, String, cBioPortal, Gepia2…).

  1. Paragraph 4.2, lane 275 “Due in part to their labile interaction with the ligase and a rapid degradation by the proteasome, 275 the identification of substrates of E3 ubiquitin ligase is rather challenging [45, 46]”. This is not true, interaction between the E3s and their substrates are not labile. Reference 45 which asserts this is not based on any reference. The authors have to rectify. A large number of substrates for many E3s have been easily identified by techniques such as co-immuno-precipitation or using yeast two-hybrid screens. However, these techniques only allow the detection of relatively strong interactions. The same is true for TRIP12 for which a significant number of substrates have already been identified (cf first paragraph of paragraph 4.2).

  1. The authors should homogenize the way of writing the name of genes and proteins considering the international Gene/Protein nomenclature guidelines for Human

- Gene symbols are italicized; all letters are in upper case ->  eg: TRIP12

- Proteins designations: same as the gene symbol, but not italicized all in upper case  ->   eg: TRIP12.

Minor points

  1. In the introduction part, in the second paragraph, the authors should clarify whether all TRIP proteins actually interact with the thyroid hormone receptor or whether the TRIP classification is simply due to sequence homologies.

  1. In the introduction part, the authors should rephrase the sentence lane 42 that is not clear: “Briefly, the first group (TRIP1 to TRIP11) needs the ligand to interact while for the second group (TRIP12 to TRIP15) the ligand inhibits the interaction to the receptor”. Which ligand? Who should interact with whom?

  1. Paragraph 2.1, lane 58 “FbxO36 gene encodes a protein that belongs to the F-box protein family.” The authors should add whether the function of the FBXO36 protein is known and whether this would be consistent with an expression co-regulated with TRIP12.

Moreover, Fbxo36 means F-box only protein 36 and not “(F box domain-Other 36 protein)”

  1. Lane 74, “The different sites of transcriptional termination are explained by the presence of two consensus polyadenylation signals (Fig. 1).” These two signals are not shown in the figure 1.

  1. Paragraph 2.2, lanes 77 to 80, the authors should specify which variant they are referencing to.

  1. Paragraph 2.3, lanes 108 and 114, the authors should explain how the location of the TRIP12 protein (which of the 14 variants?) is known when it is mentioned above that the available antibodies cannot differentiate the different isoforms.

  1. Paragraph 3.4, the authors should add more recent references regarding the IDR domain.

  1. Paragraph 4.1, - the authors should specify what is “G76” (lane 263).

                                   - the assumption “The ubiquitin ligase (E3) links together ubiquitin and substrate protein” must be corrected. In fact, only the HECT and RBR E3 ligases have a catalytic activity unlike the RING-type E3 ligases which correspond to 90% of E3 ligases.

  1. Figure 1. The AAA, polyA tail is not shown. The meaning of the yellow star is not given in the legend.

Author Response

The review by Brunet and collaborators presents an overview of what is known about the E3 ligase TRIP12 (mRNA expression, regulation, location, structure and functions). The authors have done a colossal work to bring together different information about the TRIP12 protein, that were present both in the literature and in different databases. It should be noted that as a result, this manuscript is a mix between a review and a research article.

Major points

  1. Important data are missing in the introduction part. Indeed, the authors should explain, in this section, what is the interest/importance of the TRIP12 protein and therefore the interest in studying it. They should also explain the aim of their review.

Response: The introduction was modified as requested by Reviewer 3. A paragraph that explains why it is important to study TRIP12 and that explains the aim of the review was added.

  1. The authors have to cite more often the references on which their claims are based. For instance, in Table 1, references regarding the functions of the different TRIP are missing.

Response: Table 1 has been modified accordingly to Reviewer 3’s comments. References regarding the functions of each of the TRIP proteins are now added to Table 1.

Other examples:

  • lane 76 “…pathway where the short form is more efficiently translated”
  • Response: The corresponding reference was added to the end of the sentence.
  •  
  • lane 188 “…the Super Elongation Complex (i.e.:AF9, AF4/FMR2)”
  • Response: The corresponding references were added.
  •  
  • in the paragraph 4.1 there is no reference.

Response: Missing references were added in the paragraph 4.1. as requested by Reviewer 3.

  •  
  • In the paragraph 4.2, between lanes 288 and 302 the authors have to add missing references in several sentences; for instance, lane 291 “…exception of APP-BP1 that is monoubiquitinated.”

… etc…

Response: Missing references were added in the paragraph 4.2. as requested by Reviewer 3.

The authors should also add references in the legends.

Response: The corresponding references were added to Figure 6, 7 and 8.

  1. The authors should explain what software or databases they have used and not just give their names. What does the software do or what information does it contain? (IUPred, IDEAL, InterProScan, BioGrid, String, cBioPortal, Gepia2…).

Response: To address this concern, a special paragraph “Software and databases” was added to the end of the Review.

  1. Paragraph 4.2, lane 275 “Due in part to their labile interaction with the ligase and a rapid degradation by the proteasome, 275 the identification of substrates of E3 ubiquitin ligase is rather challenging [45, 46]”. This is not true, interaction between the E3s and their substrates are not labile. Reference 45 which asserts this is not based on any reference. The authors have to rectify. A large number of substrates for many E3s have been easily identified by techniques such as co-immuno-precipitation or using yeast two-hybrid screens. However, these techniques only allow the detection of relatively strong interactions. The same is true for TRIP12 for which a significant number of substrates have already been identified (cf first paragraph of paragraph 4.2).

Response: This is a very good point. We agree that the first sentence of paragraph 4.2 is in contradiction with Table 3 that lists 76 TRIP12 interactors identified by affinity techniques and/or two-hybrid screens. Lines 291-293 have been deleted.

  1. The authors should homogenize the way of writing the name of genes and proteins considering the international Gene/Protein nomenclature guidelines for Human

- Gene symbols are italicized; all letters are in upper case ->  eg: TRIP12

- Proteins designations: same as the gene symbol, but not italicized all in upper case  ->   eg: TRIP12.

Response: The gene symbols and protein designations were modified accordingly.

Minor points

  1. In the introduction part, in the second paragraph, the authors should clarify whether all TRIP proteins actually interact with the thyroid hormone receptor or whether the TRIP classification is simply due to sequence homologies.

Response: According to Reviewer 3’s comments, the second paragraph of the introduction was rephrased to explain that “all the TRIPs interacted with Thyroid hormone Receptor”. We also now explain that “protein sequences of the 15 TRIPs are unrelated” (see also point 7).

  1. In the introduction part, the authors should rephrase the sentence lane 42 that is not clear: “Briefly, the first group (TRIP1 to TRIP11) needs the ligand to interact while for the second group (TRIP12 to TRIP15) the ligand inhibits the interaction to the receptor”. Which ligand? Who should interact with whom?

Response: We agree that this sentence was unclear. We rephrased it.

Paragraph 2.1, lane 58 “FbxO36 gene encodes a protein that belongs to the F-box protein family.” The authors should add whether the function of the FBXO36 protein is known and whether this would be consistent with an expression co-regulated with TRIP12.

Response: The sentences “Up to now, the functions of FBXO36 are unknown. In general, F-box proteins are part of SCF (SKP1, Cullin, F-box protein) ubiquitin-ligase complexes, in which they bind to substrates for ubiquitin-mediated proteolysis. This would be consistent with the fact that TRIP12 and FBXO36 expression are co-regulated.” were added in the manuscript.

Moreover, Fbxo36 means F-box only protein 36 and not “(F box domain-Other 36 protein)”

Response: The error was corrected.

  1. Lane 74, “The different sites of transcriptional termination are explained by the presence of two consensus polyadenylation signals (Fig. 1).” These two signals are not shown in the figure 1.

Response: The two polyadenylation signals are abbreviated as PAS and are indicated in the Figure 1.

  1. Paragraph 2.2, lanes 77 to 80, the authors should specify which variant they are referencing to.

Response: The corresponding variants were specified in the text.

  1. Paragraph 2.3, lanes 108 and 114, the authors should explain how the location of the TRIP12 protein (which of the 14 variants?) is known when it is mentioned above that the available antibodies cannot differentiate the different isoforms.

Response: We totally agree with your comments. This paragraph is confusing. We replaced it by the following paragraph “Unfortunately, commercially available antibodies are directed against epitopes that are common to the long isoforms of TRIP12 (except the isoform d). Therefore, they cannot discriminate the different long isoforms. However, immunostaining analysis using these antibodies reveal a nuclear localization of TRIP12 in several cell lines [11, 12] and in human tissues (Human Protein Atlas).”

  1. Paragraph 3.4, the authors should add more recent references regarding the IDR domain.

Response: The corresponding references were added to this paragraph.

  1. Paragraph 4.1, - the authors should specify what is “G76” (lane 263).

              - the assumption “The ubiquitin ligase (E3) links together ubiquitin and substrate protein” must be corrected. In fact, only the HECT and RBR E3 ligases have a catalytic activity unlike the RING-type E3 ligases which correspond to 90% of E3 ligases.

Response: The meaning of G76 is given lines 267-268 “C-terminal glycine 76 (G76) of ubiquitin”. However, it is specified again line 270-271.

We totally agree that only HECT and RBR E3 ligases have a catalytic activity. Paragraph 4.1 was corrected (lines 279, 283-284) and the particular mode of action of RING-E3s is now indicated.

Figure 1. The AAA, polyA tail is not shown. The meaning of the yellow star is not given in the legend.

Response: We are sorry but the Figure 1 has been shortened after its insertion in the manuscript during the editing process. This mistake will be corrected.

The yellow star corresponds to the cysteine-catalytic site. The meaning of the yellow star was added in the Figure 1 legend.

Round 2

Reviewer 2 Report

The authors addressed the issues raised by this reviewer. However, the reviewer is not satisfactorily convinced of the biological significance of TRIP12 as a topic for a review. Abstract and Introduction need to be improved to persuade the potential readers to appreciate the biological importance of TRIP12.

The abstract needs to summarize molecular features and pathological relevance as well as key biological functions which are briefly mentioned in the revised manuscript. Just listing topics to be covered in the main text is not helpful to guide the readers.

The introduction needs to be refined so that the readers can follow the logics of the authors on the biological significance of TRIP12 among other 14 TRIP members. Also desired would be presentation of somewhat quantitative description on how important TRIP12 research has been: for example, the total number of research papers registered in PubMed and so on.  

Author Response

We thank Reviewer 2 for its comments that will make our manuscript more appealing for the readers of IJMS.
We are convinced of the biological importance of TRIP12 that is supported by the embryonic lethality of a mouse model bearing an inactivating mutation in TRIP12 gene. Moreover, further studies demonstrated the important participation of TRIP12 in several central biological processes such as cell cycle progression, DNA damage repair, differentiation, etc… by mediating the degradation of key proteins (RNF168, BAF157, P14/ARF, etc…).
The choice of TRIP12 among the fifteen TRIP family members has been oriented by the fact that TRIP12 is the only TRIP family member that has been characterized as an E3-ubiquitin ligase. As the topic of the IJMS special issue is “The Role of E3 Ligases and Deubiquitinating Enzymes in Cellular Signaling, Diseases, and Therapeutics”, we thought legitimate to submit our manuscript for publication in this issue. To avoid any confusion on the aim of the review, the Table 1 listing the different TRIP family members was removed from the main text and placed as Supplemental Table.
As suggested, we modified the Abstract and the Introduction to better persuade the readers to appreciate the biological importance of TRIP12. The molecular features, pathological relevance and biological functions were better summarized in the Abstract.
The listing of the topics covered in the main text was deleted from the Abstract.
The number of references related to TRIP12 on PubMed was added in the Introduction.

Author Response

none